# ATLAS GAUSSIANS DIFFUSION FOR 3D GENERATION

**Haitao Yang**[1*]  **Yuan Dong**[2*]  **Hanwen Jiang**[1]  **Dejia Xu**[1]  **Georgios Pavlakos**[1]  **Qixing Huang** [1]
[1]The University of Texas at Austin      [2]Alibaba Group

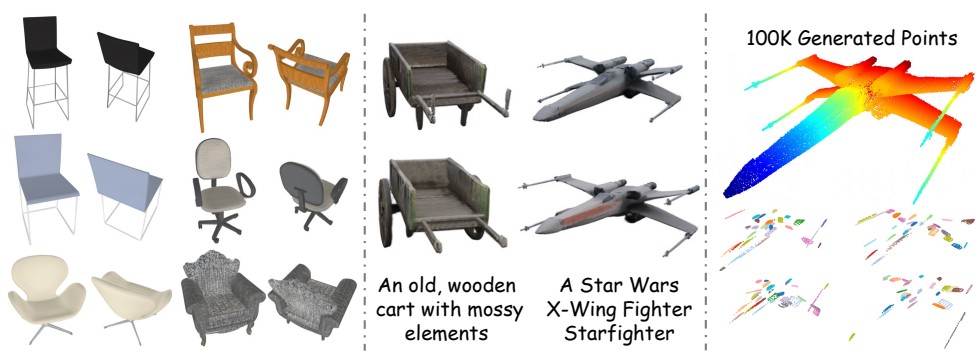

Figure 1: We propose the Atlas Gaussians representation for 3D generation. Our method supports both unconditional (left) and conditional (middle) generation with great diversity. With Atlas Gaussians, we can generate a sufficiently large, and theoretically infinite, number of 3D Gaussian points. To demonstrate this, 100K Gaussian points generated by our method are shown (right). Additionally, we iteratively sample 32 patches of the shape, displaying each set in one of four sub-figures (right).

## ABSTRACT

Using the latent diffusion model has proven effective in developing novel 3D generation techniques. To harness the latent diffusion model, a key challenge is designing a *high-fidelity* and *efficient* representation that links the latent space and the 3D space. In this paper, we introduce Atlas Gaussians, a novel representation for feed-forward native 3D generation. Atlas Gaussians represent a shape as the union of local patches, and each patch can decode 3D Gaussians. We parameterize a patch as a sequence of feature vectors and design a learnable function to decode 3D Gaussians from the feature vectors. In this process, we incorporate UV-based sampling, enabling the generation of a sufficiently large, and theoretically infinite, number of 3D Gaussian points. The large amount of 3D Gaussians enables the generation of high-quality details. Moreover, due to local awareness of the representation, the transformer-based decoding procedure operates on a patch level, ensuring efficiency. We train a variational autoencoder to learn the Atlas Gaussians representation, and then apply a latent diffusion model on its latent space for learning 3D Generation. Experiments show that our approach outperforms the prior arts of feed-forward native 3D generation. Project page: `https://yanghtr.github.io/projects/atlas_gaussians`.

## 1  INTRODUCTION

3D generation has become increasingly important in various domains, including virtual reality, gaming, and film production. Recent advances in diffusion models (Ho et al., 2020; Song et al., 2020) have improved the quality of 3D generation, offering superior performance over previous methods, such as variational autoencoders (VAEs) (Kingma & Welling, 2013; Mo et al., 2019) and generative adversarial networks (GANs)(Goodfellow et al., 2014; Gao et al., 2022).

---

* Equal Contribution

Despite progress, the effectiveness of 3D generation still falls short compared to 2D generation models. The robust performance in 2D generation is largely due to the effective integration of VAEs with latent diffusion models (LDMs) (Rombach et al., 2022). A primary challenge hindering the complete success of this paradigm in 3D generation is the development of a high-fidelity 3D representation that can be efficiently embedded into a low-dimensional latent space. Pioneering efforts have applied diffusion models to various traditional 3D representations, such as point clouds (Zeng et al., 2022; Zhou et al., 2021; Luo & Hu, 2021), meshes (Liu et al., 2023e), occupancy fields (Zheng et al., 2023; Zhang et al., 2023a) and signed distance functions (Cheng et al., 2023; Li et al., 2023b; Zhang et al., 2024). However, these approaches often focus solely on modeling geometry without considering the appearance attributes. More recently, a notable attempt (Lan et al., 2024) has designed a VAE that uses volume rendering techniques (Mildenhall et al., 2020) to incorporate appearance modeling. However, volume rendering presents inherent limitations, including slow rendering speeds and constrained rendering resolutions. To overcome these limitations, we have developed a VAE that leverages the latest 3D Gaussian representation (Kerbl et al., 2023), which significantly improves both the quality and speed of rendering.

Designing a VAE based on 3D Gaussians presents considerable challenges. The first challenge lies in creating an efficient decoder capable of mapping low-dimensional latents to 3D Gaussians. Existing methods for 3D Gaussian decoding focus predominantly on reconstruction tasks and typically lack an information bottleneck design (Tang et al., 2024; Yinghao et al., 2024), thus failing to provide low-dimensional latent directly. Alternatively, they often require multiple complex and interdependent components (Xu et al., 2024a; Zou et al., 2023). The second challenge involves generating a sufficiently large number of 3D Gaussians efficiently, since high-quality rendering necessitates an adequate quantity of these Gaussians. Some current methods (Xu et al., 2024a; Zou et al., 2023) address this by employing additional complex point-upsampling networks to increase the number of 3D Gaussians, which inherently limits the number of Gaussians that can be generated. Other techniques (Tang et al., 2024; Yinghao et al., 2024) utilize image representations to generate a large number of 3D Gaussians. However, for all these methods, more network parameters are usually required as the number of 3D Gaussians increases.

To address the challenges of designing VAEs for 3D Gaussians, we propose Atlas Gaussians, a new representation for 3D generation. This representation is inspired by surface parameterization (Floater & Hormann, 2005), a foundation technique in many graphics applications where surface attributes are sampled and stored in a 2D texture map. Specifically, Atlas Gaussians model the shape as a union of local patches, with each patch decoding 3D Gaussians via UV-based sampling. By parameterizing 3D Gaussians in the UV space, we can easily generate a sufficiently large, and theoretically infinite, number of 3D Gaussians. Unlike traditional surface parameterization approaches, the UV mapping in Atlas Gaussians is learned end-to-end. A significant advantage of Atlas Gaussians is that the sampling process does not require additional network parameters as the number of 3D Gaussians increases.

We design a transformer-based decoder to map low-dimensional latents to Atlas Gaussians. This decoder is specifically structured to disentangle geometry and appearance features, facilitating faster convergence and improved representation capabilities. Using the local awareness of Atlas Gaussians, we also reduce computational complexity by decomposing the self-attention layers. Finally, the latent space learned by our VAE can be applied to existing latent diffusion models efficiently.

Note that in contrast to the main approach in 3D generation that uses the multi-view representation (Wang & Shi, 2023; Long et al., 2023; Liu et al., 2023d; Shi et al., 2023b), our approach is inherently 3D-based. Therefore, a key advantage is that we do not need to address the challenging multi-view consistency issue associated with the multi-view representation. Moreover, the rendering module of Atlas Gaussians allows representation learning from images.

In summary, we make the following contributions.

 • We propose Atlas Gaussians, a new 3D representation that can efficiently decode a sufficiently large and theoretically infinite number of 3D Gaussians for high-quality 3D generation.

 • We design a new transformer-based decoder to efficiently map low-dimensional latents to Atlas Gaussians, using separate branches to disentangle geometry and appearance features.

 • We pioneer the integration of 3D Gaussians into the VAE + LDM paradigm, demonstrating superior performance on standard 3D generation benchmarks.

## 2 RELATED WORK

**3D representation.** 3D reconstruction and generation benefit from different 3D representations by leveraging their unique properties. These representations include explicit representations (Wu et al., 2016; Mittal et al., 2022; Ren et al., 2024; Zeng et al., 2022; Zhou et al., 2021; Luo & Hu, 2021; Sun et al., 2020a; Xie et al., 2021; Yang et al., 2019; Sun et al., 2020b; Achlioptas et al., 2018; Fan et al., 2017; Liu et al., 2023e; Nash et al., 2020; Siddiqui et al., 2023; Groueix et al., 2018; Chen et al., 2020; Kerbl et al., 2023; Tang et al., 2024; Yinghao et al., 2024; Zou et al., 2023; Szymanowicz et al., 2024) and implicit representations (Park et al., 2019; Chen & Zhang, 2019; Mescheder et al., 2019; Li et al., 2023b; Zheng et al., 2022; Shue et al., 2023; Jiang et al., 2022; Hui et al., 2022; Yan et al., 2022; Mildenhall et al., 2020; Zhang et al., 2022; Chan et al., 2022; Gu et al., 2023; Chen et al., 2023a; Cao et al., 2024; Müller et al., 2023; Watson et al., 2023). In this paper, we focus on 3D Gaussians (Kerbl et al., 2023; Tang et al., 2024; Yinghao et al., 2024; Zou et al., 2023; Szymanowicz et al., 2024; He et al., 2024), which possess high-quality rendering procedures that allow learning from image supervisions. However, existing results mainly focus on reconstructing 3D Gaussians. Our goal is to push the state-of-the-art in generative 3D Gaussians.

Our method is also related to the Atlas representation initially proposed by AtlasNet (Groueix et al., 2018) and its subsequent extensions (Deprelle et al., 2019; Liu et al., 2019; Feng et al., 2022). AtlasNet models the shape as a union of independent MLPs, thus limiting the number of patches to a few dozen. In contrast, our proposed Atlas Gaussians model each patch using a patch center and patch features. This efficient encoding allows us to generate a significantly larger number of patches, providing stronger representation capabilities. Additionally, we use a transformer to learn this representation instead of an MLP, resulting in better scalability.

**Diffusion models.** Diffusion models (Sohl-Dickstein et al., 2015; Song et al., 2021; 2020) have been dominant for diverse generation tasks, including image (Ho & Salimans, 2022; Zhang et al., 2023b; Podell et al., 2023; Rombach et al., 2022), video (Ho et al., 2022b;a; Blattmann et al., 2023), audio (Huang et al., 2023; Kong et al., 2020; Liu et al., 2023a) and text (Li et al., 2022; Gong et al., 2022; Lin et al., 2023b). The success of these models typically follows the VAE + LDM paradigm. Although pioneering efforts (Zeng et al., 2022; Zhou et al., 2021; Luo & Hu, 2021; Liu et al., 2023e; Zheng et al., 2023; Zhang et al., 2023a; Cheng et al., 2023; Li et al., 2023b; Jun & Nichol, 2023) have attempted to apply this paradigm to 3D, the problem remains unsolved and has not achieved the same level of success. We argue that one of the main reasons is the need for an efficient VAE to represent high-quality 3D content, which is the key contribution of this paper.

**3D generation.** 3D generation methods can be classified into two genres. The first is optimization-based methods (Jain et al., 2022; Sun et al., 2023; Wang et al., 2023; Lin et al., 2023a; Wang et al., 2024; Chen et al., 2023b), which are time-consuming due to per-shape optimization. For example, DreamField (Jain et al., 2022) uses CLIP guidance. DreamFusion (Poole et al., 2022) and SJC (Wang et al., 2023) introduce 2D diffusion priors in different formats. Magic3D (Lin et al., 2023a) employs a coarse-to-fine pipeline to improve convergence speed. Prolificdreamer (Wang et al., 2024) uses variational score distillation to enhance generation fidelity. Fantasia3D (Chen et al., 2023b) disentangles geometry and texture to achieve higher quality generation.

The second genre involves training a generalizable feed-forward network to output 3D content, which allows for fast 3D generation but often with less details. One direction is to apply existing generative modeling techniques (VAE, GAN, normalizing flow (Rezende & Mohamed, 2015), autoregressive model (Bengio et al., 2000; Graves, 2013) and diffusion model) directly on various 3D representations. Another direction (Xu et al., 2024b; Liu et al., 2024; 2023b;c; Shi et al., 2023a; Hong et al., 2024; Tang et al., 2024) uses 2D as intermediate representation, integrating 2D diffusion models (Ho et al., 2022a; Shi et al., 2023b). For example, Instant3D (Li et al., 2023a) trains a diffusion model to generate sparse multi-view images and uses a reconstruction model to derive the 3D shapes from the 2D images. VFusion3D (Han et al., 2024) and V3D (Chen et al., 2024) use video diffusion models to improve consistency between generated images. However, these methods rely on 2D representations and often suffer from multi-view consistency issues. In this paper, we push the state-of-the-art in the first direction by developing a novel representation of Atlas Gaussians. Our representation is efficient and allows for learning 3D generative models from images.

Figure 2: (Left) Atlas Gaussians $\mathcal{A}$ model the shape as a union of patches, where each patch can decode 3D Gaussians. (Right) Each patch $\boldsymbol{a}_i$ is parameterized by patch center $\boldsymbol{x}_i$ and patch features $\boldsymbol{f}_i$ and $\boldsymbol{h}_i$. The 3D Gaussians are decoded via the UV-based sampling.

## 3 METHOD

In this section, we first introduce our Atlas Gaussians representation (Sec. 3.1). Then we introduce how we learn a VAE to connect the 3D space with the latent space (Sec. 3.2). Finally, we introduce how we learn the generative model using latent diffusion in the learned latent space (Sec. 3.3).

### 3.1 ATLAS GAUSSIANS REPRESENTATION

In 3D Gaussian Splatting (Kerbl et al., 2023), each 3D Gaussian $\mathbf{g}$ can be parameterized with a center $\boldsymbol{\mu} \in \mathbb{R}^3$, scale $\mathbf{s} \in \mathbb{R}^3$, rotation quaternion $\mathbf{r} \in \mathbb{R}^4$, opacity $\mathrm{o} \in \mathbb{R}$ and color $\mathbf{c} \in \mathbb{R}^3$. To achieve high-quality rendering results, typically a sufficiently large number of 3D Gaussians are required.

The key idea of Atlas Gaussians is to represent the shape as a union of $M$ local patches $\mathcal{A} = \{\boldsymbol{a}_i\}_{i=1}^M$, where each patch $\boldsymbol{a}_i$ can decode 3D Gaussians through UV-based sampling. As shown in Figure 2, we parameterize each local patch $\boldsymbol{a}_i := (\boldsymbol{x}_i, \boldsymbol{f}_i, \boldsymbol{h}_i)$ with patch center $\boldsymbol{x}_i \in \mathbb{R}^3$, the geometry features $\boldsymbol{f}_i \in \mathbb{R}^{4 \times d}$ and the appearance features $\boldsymbol{h}_i \in \mathbb{R}^{4 \times d}$. More specifically, we parameterize the geometry and appearance features as the features at the four corners of the local patch in the UV space. We denote $\boldsymbol{f}_i = (\boldsymbol{f}_{i1}, \boldsymbol{f}_{i2}, \boldsymbol{f}_{i3}, \boldsymbol{f}_{i4})$ and $\boldsymbol{h}_i = (\boldsymbol{h}_{i1}, \boldsymbol{h}_{i2}, \boldsymbol{h}_{i3}, \boldsymbol{h}_{i4})$. This type of feature disentanglement can facilitate more effective learning (Gao et al., 2022; Zou et al., 2023; Xu et al., 2024a). We use the feature $\boldsymbol{f}_i$ to decode Gaussian positions while using $\boldsymbol{h}_i$ to decode the rest of Gaussian attributes. We also assign a 2D coordinate $\boldsymbol{u}_{ij} \in \mathbb{R}^2$ to each feature vector $\boldsymbol{f}_{ij}$ and $\boldsymbol{h}_{ij}$ as a positional embedding, where $\boldsymbol{u}_{i1} = (0,0)$, $\boldsymbol{u}_{i2} = (1,0)$, $\boldsymbol{u}_{i3} = (1,1)$, $\boldsymbol{u}_{i4} = (0,1)$, $\forall i$.

Note that this representation is motivated by the concept of UV-map, in which the four corners describe the corners of the rectangular parameter domain. As we shall discuss later, the features $\boldsymbol{f}_i$ and $\boldsymbol{h}_i$ are learned end-to-end with 3D Gaussians. This approach takes advantage of end-to-end learning, while the specific network design promotes learning better features.

For generation, we randomly sample query points in the predefined unit square UV space for each patch $\boldsymbol{a}_i$. Each point is then decoded into a 3D Gaussian. Given a query point $\boldsymbol{q}_{ij} \in [0,1]^2$, we map the 2D coordinate $\boldsymbol{q}_{ij}$ to the center of 3D Gaussians as:

$$\boldsymbol{\mu}_{ij} = \phi(\boldsymbol{q}_{ij}, \boldsymbol{u}_i, \boldsymbol{f}_i), \tag{1}$$

where $\phi$ is the mapping function, which takes the query point location $\boldsymbol{q}_{ij}$, the predefined location of patch feature vectors $\boldsymbol{u}_i$, and the geometry features $\boldsymbol{f}_i$ as inputs. We implement the mapping function using interpolation in the 2D space:

$$\boldsymbol{\mu}_{ij} = \mathrm{MLP}(\sum_{k=1}^4 w(\boldsymbol{q}_{ij}, \boldsymbol{u}_{ik}, \boldsymbol{f}_{ik}) \cdot \boldsymbol{f}_{ik}) + \boldsymbol{x}_i, \tag{2}$$

where $w$ is the weight function of the four corners and we use an MLP to decode the residual between the Gaussian location and the patch center $\boldsymbol{x}_i$

One design choice for $w$ is the bilinear interpolation weight function, which has been widely used in feature decoding (Müller et al., 2022). However, these linear weights purely based on coordinates have limited representation ability. Inspired by (Zhang et al., 2023a), we design a more powerful

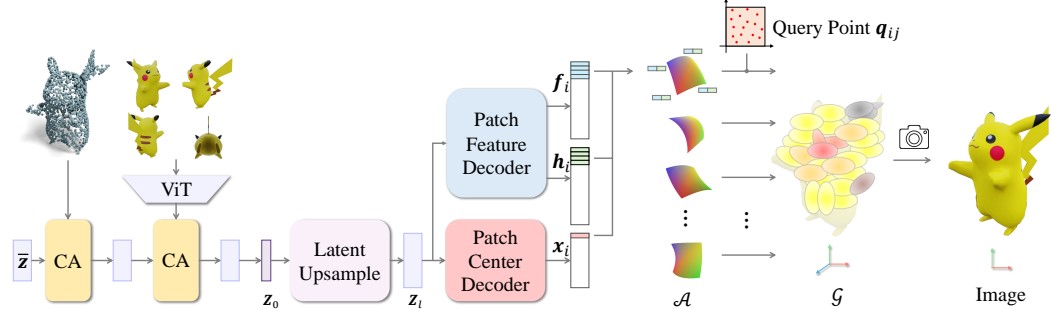

Figure 3: The proposed VAE architecture. CA denotes the cross-attention layer. For simplicity, the variational component of the VAE is omitted. The latent $z_0$ is used for latent diffusion.

weight function defined on both coordinates and features:

$$w(\boldsymbol{q}_{ij}, \boldsymbol{u}_{ik}, \boldsymbol{f}_{ik}) = \frac{\widetilde{w}(\boldsymbol{q}_{ij}, \boldsymbol{u}_{ik}, \boldsymbol{f}_{ik})}{\sum_{l=1}^{4} \widetilde{w}(\boldsymbol{q}_{ij}, \boldsymbol{u}_{il}, \boldsymbol{f}_{il})}, \tag{3}$$

$$\text{and } \widetilde{w}(\boldsymbol{q}_{ij}, \boldsymbol{u}_{il}, \boldsymbol{f}_{il}) = e^{\omega_2(\boldsymbol{q}_{ij})^T (\boldsymbol{f}_{il} + \omega_2(\boldsymbol{u}_{il}))/\sqrt{d}}, \tag{4}$$

where $\omega_2 : \mathbb{R}^2 \mapsto \mathbb{R}^d$ is the sinusoidal positional encoding function with MLP projection.

Similarly, the remaining Gaussian attributes are decoded by the function $\psi$:

$$(\mathbf{s}_{ij}, \mathbf{r}_{ij}, \mathbf{o}_{ij}, \mathbf{c}_{ij}) = \psi(\boldsymbol{q}_{ij}, \boldsymbol{u}_i, \boldsymbol{h}_i) = \text{MLP}(\sum_{k=1}^{4} w(\boldsymbol{q}_{ij}, \boldsymbol{u}_{ik}, \boldsymbol{h}_{ik}) \cdot \boldsymbol{h}_{ik}), \tag{5}$$

The benefits of Atlas Gaussians representation are three-fold. First, through UV-based sampling in a unit square, Atlas Gaussians enable easy generation of a sufficiently large number of 3D Gaussians. They also possess the potential to generate a variable and theoretically infinite number of 3D Gaussians. Second, Atlas Gaussians utilize a non-linear learnable weight function based on the MLP projection of the positional encoding, which has a stronger representation ability than the existing linear interpolation weight. Third, Atlas Gaussians are computation efficient with low memory overhead. Another important property of Atlas Gaussians is that they do not require extra network parameters when scaling up the number of generated 3D Gaussians. We also provide an ablation study in Section 4.4 to validate the nice properties of Atlas Gaussians.

## 3.2 STAGE 1: VAE

We design a VAE to link the latent space and the 3D space. The overall pipeline is shown in Figure 3.

**Encoder.** Following the latent set representation in (Zhang et al., 2023a), the encoder takes shape information as input and outputs a latent set $\boldsymbol{z}_0 \in \mathbb{R}^{n \times d_0}$, where $n$ is the size of the set in the latent set representation and $d_0$ is the latent dimension. The shape information includes point cloud $\mathcal{P} = \{\boldsymbol{p}_i \in \mathbb{R}^3\}$ and sparse view RGB images $\mathcal{I} = \{\boldsymbol{I}_i \in \mathbb{R}^{H \times W \times 3}\}$. We encode the location of points into positional embeddings, resulting in features in $\mathbb{R}^{|\mathcal{P}| \times d}$. Meanwhile, a ViT network (Dosovitskiy et al., 2021; Oquab et al., 2024) embeds each image into features of shape $\mathbb{R}^{h \times w \times d}$, resulting in the features of all images in $\mathbb{R}^{(|\boldsymbol{I}| \times h \times w) \times d}$. We then initialize the latent features $\bar{\boldsymbol{z}} \in \mathbb{R}^{n \times d}$ as the encoding of the points sampled using the farthest point sampling (Zhang et al., 2023a) and use transformers with cross-attention to aggregate input shape information. After feature aggregation, we use an MLP to map the features to a lower-dimensional space $\mathbb{R}^{n \times d} \to \mathbb{R}^{n \times d_0}$, facilitating efficient latent diffusion. The process is summarized as follows:

$$\boldsymbol{z}' = \text{CrossAttn}(\bar{\boldsymbol{z}}, \mathcal{P}), \quad \boldsymbol{z}'' = \text{CrossAttn}(\boldsymbol{z}', \mathcal{I}), \quad \boldsymbol{z}''' = \text{SelfAttn}(\boldsymbol{z}''), \quad \boldsymbol{z}_0 = \text{MLP}(\boldsymbol{z}'''). \tag{6}$$

In this notation, we omit the feed forward MLP of the transformers for simplicity. Intuitively, the query $\bar{\boldsymbol{z}}$ is updated by aggregating the input point features and image features.

Note that attention is invariant to the permutation of features. We also avoid pose-dependent operations (Jiang et al., 2024; Jun & Nichol, 2023) in our design to enhance generality and extendability.

**Decoder.** The decoder recovers patch features $\mathcal{A} = \{(\boldsymbol{x}_i, \boldsymbol{f}_i, \boldsymbol{h}_i)\}_{i=1}^{M}$ from the latent code $\boldsymbol{z}_0$. In the decoder, we upsample the latent code, decode the patch centers $\{\boldsymbol{x}_i\}_{i=1}^{M}$, and then decode the patch geometry and appearance features $\{(\boldsymbol{f}_i, \boldsymbol{h}_i)\}_{i=1}^{M}$.

The upsampling module is designed to increase both the length and the feature dimension of the latent code, from $\boldsymbol{z}_0 \in \mathbb{R}^{n \times d_0}$ to $\boldsymbol{z}_l \in \mathbb{R}^{M \times d}$. In detail, we first use a learnable query $\boldsymbol{y}$ to aggregate information from the latent code $\boldsymbol{z}_0$ using transformers with cross-attention. Then, we increase its channel dimension to $d$ using self-attention transformers, leading to the characteristic $\boldsymbol{z}_1 \in \mathbb{R}^{n \times d}$. We then use an MLP to increase the feature dimension of $\boldsymbol{z}_1$ into $\frac{M}{n}d$, leading to latent features in $\mathbb{R}^{n \times \frac{M}{n}d}$. We reshape the features into $\boldsymbol{z}_2 \in \mathbb{R}^{M \times d}$, which is a pixel shuffle process (Shi et al., 2016; Xu et al., 2024a). We then use another set of transformers with self-attention to obtain the output latent features $\boldsymbol{z}_l$. This process is summarized as follows:

$$\boldsymbol{z}_1 = \text{SelfAttn}(\text{CrossAttn}(\boldsymbol{y}, \boldsymbol{z}_0)), \quad \boldsymbol{z}_2 = \text{PixelShuffle}(\text{MLP}(\boldsymbol{z}_1)), \quad \boldsymbol{z}_l = \text{SelfAttn}(\boldsymbol{z}_2), \quad (7)$$

where $\boldsymbol{y}$ is initialized using a Gaussian distribution. Again, we omit the feed-forward MLP in the transformers for simplicity. We then decode the patch centers using transformers with self-attention:

$$\{\boldsymbol{x}_i\}_{i=1}^{M} = \text{SelfAttn}(\boldsymbol{z}_l). \quad (8)$$

We use a two-branch module to decode geometry and appearance features $\{(\boldsymbol{f}_i, \boldsymbol{h}_i)\}_{i=1}^{M}$ from upsampled latent code $\boldsymbol{z}_l$. We demonstrate its architecture in Figure 4. Take the branch for decoding the geometry features $\{\boldsymbol{f}_i\}_{i=1}^{M}$ as an example. We use another upsampling module to map $\boldsymbol{z}_l \in \mathbb{R}^{M \times d}$ to $\boldsymbol{z}_f \in \mathbb{R}^{M \times \beta \times d}$, where $\beta = 4$, corresponding to the geometry features of the four corners for each patch. We then use transformers with self-attention to refine the upsampled features to get $\boldsymbol{f}_i$.

Specifically, we apply computational decomposition to the self-attention layers, as naive self-attention leads to a $\mathcal{O}(\beta^2 M^2 d)$ complexity due to the long sequence length of $M\beta$. We apply local self-attention to the features that belong to each local patch, reducing the complexity to $\mathcal{O}(\beta^2 M d)$. The features of different local patches are updated independently. This design ensures local awareness of Atlas Gaussians decoding. To further ensure global awareness, we repeat and add the global features to the local patch features.

After obtaining the Atlas Gaussians $\mathcal{A} = \{(\boldsymbol{x}_i, \boldsymbol{f}_i, \boldsymbol{h}_i)\}_{i=1}^{M}$ from the VAE, we can decode 3D Gaussians following Eq. 2 and Eq. 5.

**Training.** Similar to existing methods (Lan et al., 2024; Xu et al., 2024a; Zou et al., 2023), we utilize a 3D dataset for supervision. We first regularize the patch center to adhere to the 3D surface geometry:

$$\mathcal{L}_{\text{center}} = \mathcal{L}_{\text{CD}}(\{\boldsymbol{x}_i\}_{i=1}^{M}, \mathcal{P}_{\text{GT}}) + \mathcal{L}_{\text{EMD}}(\{\boldsymbol{x}_i\}_{i=1}^{M}, \mathcal{P}_{\text{GT}}), \quad (9)$$

where $\mathcal{P}_{\text{GT}}$ is the ground truth surface point cloud, $\mathcal{L}_{\text{CD}}$ and $\mathcal{L}_{\text{EMD}}$ are Chamfer Distance (CD) and Earth Mover's Distance (EMD), respectively. Similarly, we also supervise the centers of all 3D Gaussians with

$$\mathcal{L}_{\boldsymbol{\mu}}(S) = \mathcal{L}_{\text{CD}}(\{\boldsymbol{\mu}_{ij}\}_{i=1,j=1}^{M,S}, \mathcal{P}_{\text{GT}}) + \mathcal{L}_{\text{EMD}}(\{\boldsymbol{\mu}_{ij}\}_{i=1,j=1}^{M,S}, \mathcal{P}_{\text{GT}}), \quad (10)$$

where $\{\boldsymbol{\mu}_{ij}\}_{i=1,j=1}^{M,S}$ represents the generated point cloud by sampling $S$ points in each of the $M$ patches. When computing the loss for supervision, we can generate point clouds with different resolutions by varying $S$, thanks to Atlas Gaussians' ability to dynamically generate a variable number of 3D Gaussians. Note that patches may overlap, similar to AtlasNet. $\mathcal{L}_{\text{center}}$ and $\mathcal{L}_{\boldsymbol{\mu}}(S)$ encourage the patches to distribute more uniformly across the surface, thereby making better use of the 3D Gaussians.

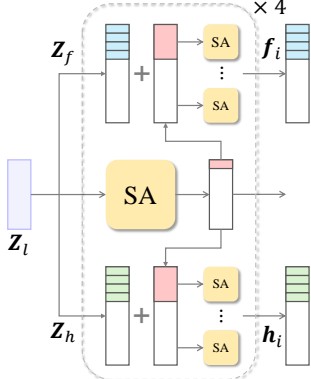

Figure 4: In the patch feature decoder, global features are broadcast and added to the local features. SA denotes the self-attention layers.

While surface points are independent samples of the continuous geometry, 3D Gaussians are interdependent because one 3D Gaussian can affect the attributes of its neighbors during alpha blending. To ensure consistency and achieve deterministic results, during rendering we employ a uniform $\alpha \times \alpha$ grid sampling in the UV space instead of random sampling, resulting in $N = \alpha^2 M$ 3D Gaussians. We use the differentiable renderer from (Kerbl et al., 2023) to render $V$ views of RGB, alpha, and depth images, which is supervised by mean square error:

$$\mathcal{L}_{\text{render}} = \mathcal{L}_{\text{MSE}}(\hat{\boldsymbol{I}}_{\text{rgb}}, \boldsymbol{I}_{\text{rgb}}) + \mathcal{L}_{\text{MSE}}(\hat{\boldsymbol{I}}_{\text{alpha}}, \boldsymbol{I}_{\text{alpha}}) + \mathcal{L}_{\text{MSE}}(\hat{\boldsymbol{I}}_{\text{depth}}, \boldsymbol{I}_{\text{depth}}), \quad (11)$$

where $\hat{I}_{\text{rgb}}$, $\hat{I}_{\text{alpha}}$, and $\hat{I}_{\text{depth}}$ are the predictions, $I_{\text{rgb}}$, $I_{\text{alpha}}$, and $I_{\text{depth}}$ are the ground truths. To improve the visual fidelity, we also employ the LPIPS loss (Zhang et al., 2018; Tang et al., 2024) $\mathcal{L}_{\text{LPIPS}}(\hat{I}_{\text{rgb}}, I_{\text{rgb}})$. The total loss function for the VAE is:

$$\mathcal{L}_{\text{total}} = \mathcal{L}_{\text{center}} + \sum_{S=\{1,4\}} \mathcal{L}_{\boldsymbol{\mu}}(S) + \lambda_r \left( \mathcal{L}_{\text{render}} + \mathcal{L}_{\text{LPIPS}} \right) + \lambda_{KL} \mathcal{L}_{\text{KL}}(\boldsymbol{z}), \tag{12}$$

where $\mathcal{L}_{\text{KL}}$ is the Kullback–Leibler divergence, $\lambda_r$ and $\lambda_{KL}$ are the loss weights.

### 3.3 STAGE 2: LATENT DIFFUSION MODEL

The proposed VAE provides a low-dimensional latent code that can be mapped to Atlas Gaussians. We employ EDM (Karras et al., 2022) for latent diffusion. Our denoising network follows the same architecture as (Zhang et al., 2023a), which consists of a series of transformer blocks:

$$\boldsymbol{z}^{(i)} = \text{CrossAttn}\left(\text{SelfAttn}(\boldsymbol{z}^{(i-1)}), \mathcal{C}\right), \quad i = 1, \ldots, l \tag{13}$$

where $\boldsymbol{z}^{(0)} = \boldsymbol{z}_0$ is the initial input to the network, $l$ is the number of blocks. In unconditional generation, we designate $\mathcal{C}$ as a learnable parameter. For text-conditioned generation, $\mathcal{C}$ is set as the CLIP embedding (Radford et al., 2021) of the input text prompts. Since the LDM network design is not our main contribution, we provide the implementation details in Appendix A.3.

## 4 EXPERIMENTS

We first introduce our experimental setup, and then show the results for both unconditional and conditional generation.

### 4.1 EXPERIMENTAL SETUP

Following most existing methods (Gao et al., 2022; Müller et al., 2023; Chen et al., 2023a; Lan et al., 2024), we benchmark unconditional single-category 3D generation on ShapeNet (Chang et al., 2015). We use the training split from SRN (Sitzmann et al., 2019), which comprises 4612, 2151, and 3033 shapes in the categories Chair, Car, and Plane, respectively. We render 76 views for each shape using Blender (Community, 2018) with the same intrinsic matrix as (Cao et al., 2024; Lan et al., 2024). Fréchet Inception Distance (FID@50K) and Kernel Inception Distance (KID@50K) are used for evaluation. In addition, we experiment with text-conditioned 3D generation on Objaverse (Deitke et al., 2022). We use the renderings from G-buffer Objaverse (Qiu et al., 2023) and the captions from Cap3D (Luo et al., 2023). Due to limited computational resources, we select a high-quality subset with around 18K 3D shapes. We use CLIP score (Radford et al., 2021; Hessel et al., 2021), FID and KID for evaluation. Please refer to Appendix A.1 for more implementation details.

### 4.2 UNCONDITIONAL 3D GENERATION

Table 1 presents the quantitative comparison between our method and baseline approaches. Our method outperforms all baseline approaches, including EG3D (Chan et al., 2022), GET3D (Gao et al., 2022), DiffRF (Müller et al., 2023), RenderDiffusion (Anciukevičius et al., 2023), SSDNeRF (Chen et al., 2023a) and LN3Diff (Lan et al., 2024). We also include the qualitative comparison with LN3Diff in Figure 5. Our results demonstrate significant improvements over LN3Diff, particularly in the ShapeNet Chair category, which features greater geometric, structural, and textural complexity. The results highlight the robustness and efficacy of our approach.

### 4.3 CONDITIONAL 3D GENERATION

We evaluate our method and baseline approaches (He et al., 2024; Lan et al., 2024; Tang et al., 2024; Jun & Nichol, 2023) on text-conditioned 3D generation using Objaverse. The qualitative results are presented in 6, where all text prompts are sourced from the original baseline papers. As shown in Figure 6, the image-based generalizable 3D reconstruction method (Tang et al., 2024) sometimes produces shapes with significant artifacts, primarily due to its reliance on a multi-view

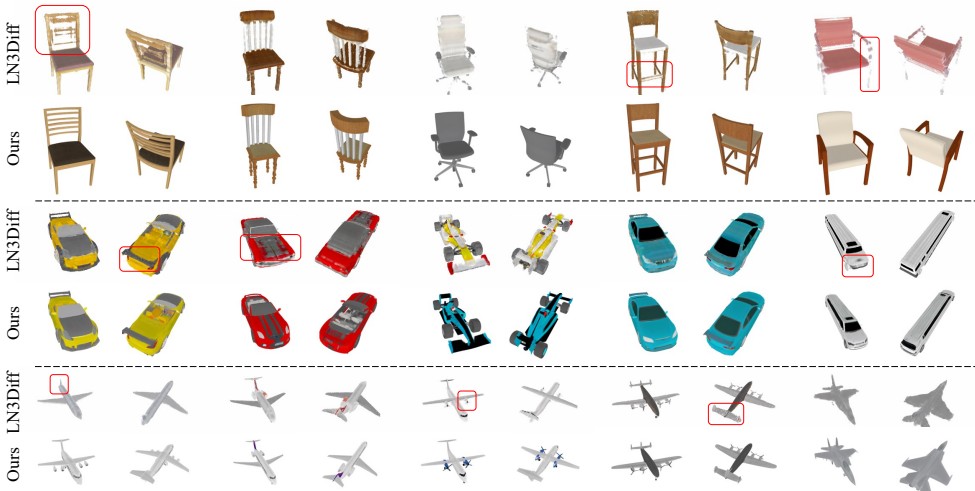

Figure 5: The comparison between LN3Diff and our method on three ShapeNet categories.

Table 1: Evaluation of single-category unconditional generation on ShapeNet.

| Method | Chair | | Car | | Plane | |
|---|---|---|---|---|---|---|
| | FID@50K | KID@50K(%) | FID@50K | KID@50K(%) | FID@50K | KID@50K(%) |
| EG3D (Chan et al., 2022) | 26.09 | 1.1 | 33.33 | 1.4 | 14.47 | 0.5 |
| GET3D (Gao et al., 2022) | 35.33 | 1.5 | 41.41 | 1.8 | 26.80 | 1.7 |
| DiffRF (Müller et al., 2023) | 99.37 | 4.9 | 75.09 | 5.1 | 101.79 | 6.5 |
| RenderDiffusion (Anciukevičius et al., 2023) | 53.30 | 6.4 | 46.50 | 4.1 | 43.50 | 5.9 |
| SSDNeRF (Chen et al., 2023a) | 65.04 | 3.0 | 47.72 | 2.8 | 21.01 | 1.0 |
| LN3Diff (Lan et al., 2024) | 16.90 | 0.47 | 17.60 | 0.49 | 8.84 | 0.36 |
| Ours | **9.90** | **0.35** | **12.15** | **0.45** | **8.09** | **0.21** |

generation network, which frequently suffers from multi-view inconsistency. In contrast, our method learns directly from 3D data, enabling the generation of consistent novel views. Additionally, our method produces higher-quality 3D shapes compared to Shap-E and LN3Diff, both of which use the VAE + LDM paradigm. However, these methods depend on volume rendering during training, which is typically restricted to low resolution. Our approach instead leverages the more efficient 3D Gaussian representation. Compared to GVGEN, our generated shapes exhibit finer details, owing to the proposed Atlas Gaussians representation, which is capable of decoding a significantly larger number of 3D Gaussians. Quantitative results are provided in Table 2, where our method achieves the best performance across all metrics and has the shortest inference time. This improvement is due to our efficient decoder design, which effectively links the low-dimensional latent space with 3D space. Additional results can be found in Appendix B.

In Figure 7, we provide a detailed analysis of our method. In Figure 7 (Left), we present text-conditioned generation results with different random seeds, demonstrating that our method produces highly diverse outputs. In Figure 7 (Right), we show that our model can be robustly controlled using different text prompts. The generated results are also significantly different from their nearest neighbors in the training dataset, highlighting the model's ability to generate novel content.

Table 2: Evaluation of text-conditioned 3D generation on Objaverse.

| | GVGEN (He et al., 2024) | LN3Diff (Lan et al., 2024) | LGM (Tang et al., 2024) | Shap-E (Jun & Nichol, 2023) | Ours |
|---|---|---|---|---|---|
| CLIP Score (ViT-B/32) ↑ | 27.33 | 27.21 | 29.62 | 30.22 | **30.66** |
| FID@6K ↓ | 132.4 | 123.8 | 117.0 | 114.5 | **109.5** |
| KID@6K (%) ↓ | 6.04 | 4.53 | 4.68 | 4.38 | **4.04** |
| Inference Time (GPU) ↓ | 28 s (V100) | 7.5 s (V100) | 6 s (TITAN V) | 33 s (TITAN V) | 4 s (TITAN V) |

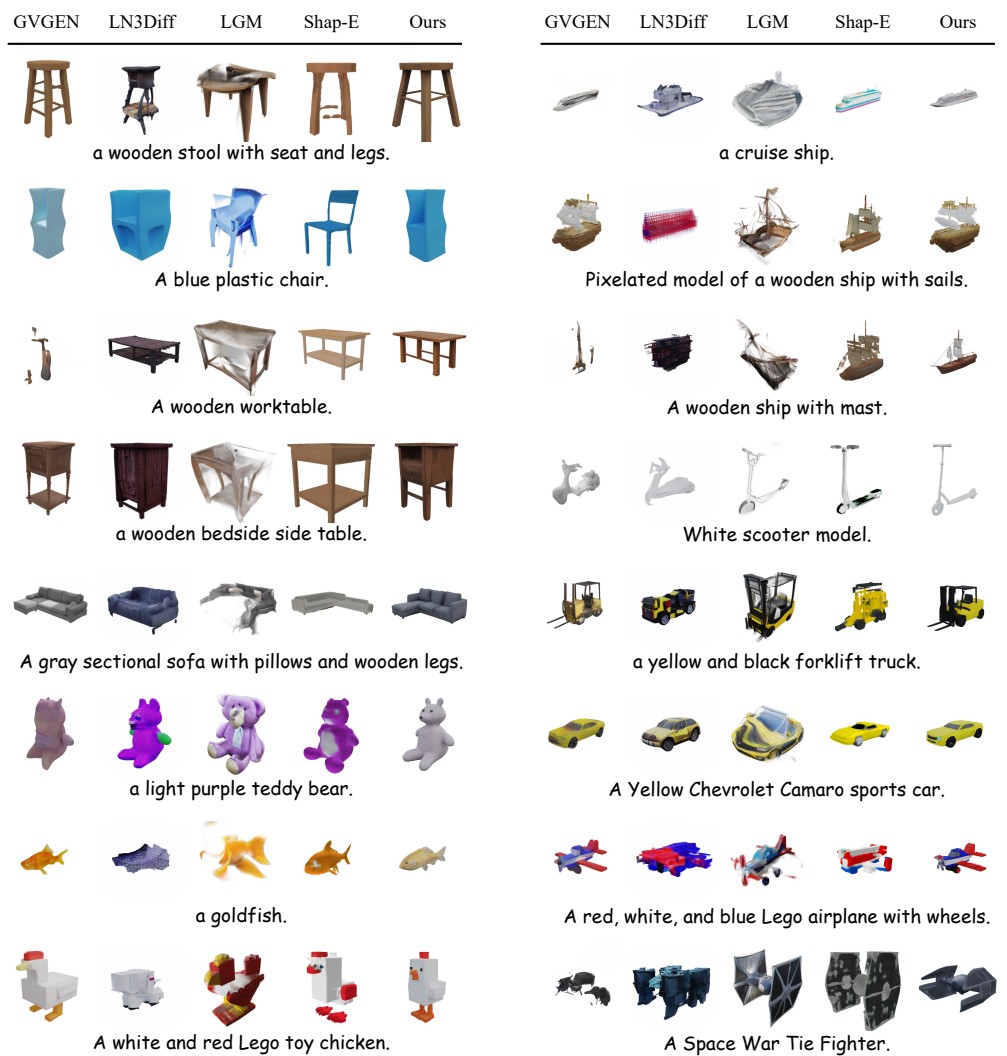

Figure 6: Comparison of text-conditioned 3D generation on Objaverse. From left to right: GV-GEN (He et al., 2024), LN3Diff (Lan et al., 2024), LGM (Tang et al., 2024), Shap-E (Jun & Nichol, 2023), and our method. All text prompts are sourced from the original baseline papers.

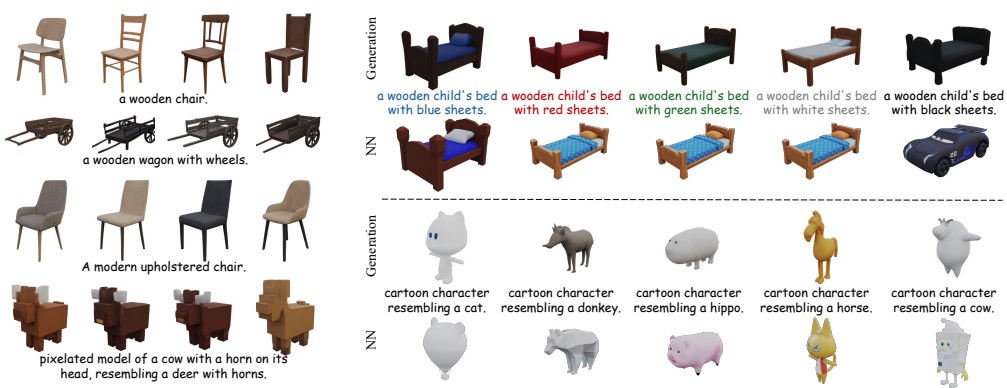

Figure 7: (Left) Our generated results demonstrate significant diversity. (Right) Our generated results align closely with the text prompts, allowing for strong controllability. In the second row of each group, we present the nearest neighbors (NN) from the training dataset.

Table 3: Ablation study on the number of 3DGS and the number of patches on Objaverse. (LPIPS↓ / MEM↓)

| #patches | #3DGS≈8K | #3DGS≈32K | #3DGS≈100K |
|---|---|---|---|
| 512 | – | – | 0.068 / 1.2G |
| 1024 | – | – | 0.060 / 1.3G |
| 2048 | 0.072 / 1.3G | 0.063 / 1.4G | 0.058 / 1.7G |
| 4096 | – | 0.061 / 2.0G | 0.057 / 2.3G |

Table 4: Ablation study on network design evaluated on ShapeNet with PSNR.

| Atlas Gaussians (Full) | 26.56 |
|---|---|
| (a) linear weights | 26.13 |
| (b) no disentangle | 25.77 |
| (c) no global feature | 26.18 |
| (d) decode $\mathbf{s},\mathbf{r},$o from $\boldsymbol{f}_i$ | 26.55 |

## 4.4 ABLATION STUDY

This section presents an ablation study on different components of our network.

**Number of 3D Gaussians and number of patches.** In Table 3, we analyze the effects of both the number of patches and the total number of 3D Gaussians on LPIPS and memory usage (MEM, which represents the additional GPU memory required for each unit increase in batch size). The results indicate that increasing either the number of patches or the number of 3D Gaussians improves LPIPS. When the number of 3D Gaussians is fixed (32k or 100k), increasing the number of patches from 2,048 to 4,096 results in a 0.1-0.2 improvement in LPIPS. However, when the number of patches is fixed (2,048 or 4,096), increasing the number of 3D Gaussians from 32k to 100k leads to a more significant improvement in LPIPS while requiring less additional GPU memory. Importantly, the network parameters remain unchanged despite the increase in 3D Gaussians. This comparison demonstrates that increasing the number of 3D Gaussians through Atlas Gaussians decoding is more efficient and effective than increasing the number of patches. Notably, the Atlas Gaussians representation combines explicit patch generation with implicit interpolation to generate 3D Gaussians, offering a balanced trade-off between compact implicit and fast explicit representations.

**Learned weights for feature decoding.** In Table 4, we ablate the VAE network design on the ShapeNet validation set using the PSNR metric. The table shows that replacing our learned weight function with bilinear interpolation weights leads to a performance drop for the VAE. This indicates that our learned nonlinear weights have stronger representation capabilities than linear weights.

**Disentangle geometry and texture features.** Atlas Gaussians use separate branches to learn geometry and appearance features. To validate the effectiveness of this design, we experimented with an alternative network version where a single set of features is shared for both geometry and appearance. As shown in Table 4, performance decreases considerably with this shared feature approach. This shows that our design facilitates more effective learning. Note that both the geometry and appearance features are generated from the shared latent $\boldsymbol{z}_0$, and disentanglement is performed during feature generation rather than in the latent space.

**Using global features.** As shown in Table 4, removing the global feature from $\boldsymbol{z}_l$ in the patch feature decoder leads to a performance drop. This outcome is expected, as the global features provide essential context that complements the local features.

**Parameter decoding scheme.** In Gaussian Splatting, opacity, scale, and rotation also influence the geometry. We experimented with decoding opacity, scale, and rotation from the geometry features and found that the performance is nearly identical to when these attributes are decoded from the appearance features. This result indicates that the key factor is decoding the Gaussian centers and RGB colors using distinct, separate branches.

## 5 CONCLUSION

This paper introduces Atlas Gaussians, a new representation for feed-forward 3D generation. Atlas Gaussians enable the efficient generation of a sufficiently large and theoretically infinite number of 3D Gaussians, which is crucial for high-quality 3D generation. We also designed a transformer-based decoder to link the low-dimensional latent space with the 3D space. With these innovations, we pioneer the integration of 3D Gaussians into the VAE + LDM paradigm, achieving superior performance and producing high-fidelity 3D content.

**Acknowledgment.** We thank Yushi Lan for his invaluable assistance in the evaluation. Qixing Huang acknowledges the support from NSF IIS-2047677, IIS-2413161, and GIFTs from Adobe and Google.

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

# A  Implementation details

## A.1  Additional details of the dataset

We adapted the training setup from LN3Diff (Lan et al., 2024), which initially used a subset of 35K shapes. These shapes cover three general categories: Transportation, Furniture, and Animals, derived from the G-buffer Objaverse (Qiu et al., 2023). Due to limited computational resources, we cleaned the dataset by filtering out duplicate and low-quality shapes (e.g., those with flat ground or poor geometry), resulting in a final subset of approximately 18K shapes for training. Given the relatively small size of our dataset, we manually align the 3D shapes to establish a consistent canonical orientation across each category, facilitating easier learning. It is important to note that baselines like LN3Diff utilize additional data from other categories, which provides them an advantage.

We randomly selected 250 text prompts for evaluation, ensuring that each testing prompt differs from the training data. For each object, we rendered 24 views uniformly, following the G-buffer Objaverse protocol, resulting in a total of 6K images. FID and KID are computed based on the 2D image feature space.

## A.2  Additional details of VAE

The VAE training consists of two stages. In the first stage, $\lambda_r$ in Eq. 12 is set to 0. Rendering occurs only in the second stage with $\lambda_r$ set to 1. In both stages, $\lambda_{KL}$ maintains $1e^{-4}$. As shown in Eq. 12, all other loss weights are set to 1 for simplicity. Our method is robust to hyperparameters; doubling or halving $\lambda_r$ in Eq. 12 results in almost identical loss curves.

We utilize a sparse point cloud of 2048 points and 4 input views, each of size $224 \times 224$. $M = 2048$ patches are used in all experiments. For the latent, $n = 512$, $d = 512$. We set $d_0$ to 4 for ShapeNet and $d_0$ to 16 in Objaverse. In ShapeNet, $\alpha$ is set to 4, resulting in $N = 32768$ 3D Gaussians. In Objaverse, $\alpha = 7$. Due to the real-time rendering capabilities of the 3D Gaussians, we are able to produce an output image of size $512 \times 512$. We use the EMD implementation from (Liu et al., 2019), which supports fast and efficient computation even with up to 8192 points. All networks are trained on 8 Tesla V100 GPUs for 1000 epochs using the AdamW optimizer (Loshchilov & Hutter, 2019)

---

The authors recently scaled up their model with more data, and we use this updated version for comparison.

with the one-cycle policy. Our VAE is trained using mixed precision (fp16) and supports a batch size of 8 per GPU. For instance, when trained on the ShapeNet dataset, it requires approximately 22GB of memory per GPU, making it accessible to a wide range of laboratories. Notably, the model does not require the latest A100 GPUs for training. Furthermore, the training process is completed in less than 30 hours. For Objaverse, the VAE training takes about 6 days.

In terms of network parameters, our model increases the number of Gaussians without adding extra network parameters. Specifically, for Objaverse, our model requires only 142M network parameters to generate 100K Gaussians, while LGM (Tang et al., 2024) requires 415M parameters for generating 64K Gaussians. This demonstrates that our representation is more efficient than purely explicit representations like LGM, offering a nice trade-off between compact implicit and fast explicit representations.

### A.3 ADDITIONAL DETAILS OF LDM

We adopt the EDM (Karras et al., 2022) framework for latent diffusion. EDM aims to learn a denoising network $D_\theta(z; \sigma, \mathcal{C})$ to convert the Gaussian distribution to the empirical distribution $p_{\text{data}}$ defined by $z$, where $\theta$ is the network parameters, $\sigma$ is the noise level sampled from a predefined distribution $p_{\text{train}}$, and $\mathcal{C}$ denotes the optional condition. $D_\theta(z; \sigma, \mathcal{C})$ is parameterized using a $\sigma$-dependent skip connection:

$$D_\theta(z; \sigma, \mathcal{C}) = c_{\text{skip}}(\sigma)z + c_{\text{out}}(\sigma)F_\theta\left(c_{\text{in}}(\sigma)z; c_{\text{noise}}(\sigma), \mathcal{C}\right), \tag{14}$$

where $F_\theta$ is the network to be trained. The training objective is

$$\mathbb{E}_{\sigma, z, n}\lambda(\sigma)||D_\theta(z; \sigma, \mathcal{C}) - z||^2, \tag{15}$$

where $\sigma \sim p_{\text{train}}, z \sim p_{\text{data}}, n \sim \mathcal{N}(0, \sigma^2 I)$. Readers interested in details on $c_{\text{skip}}(\sigma)$, $c_{\text{out}}(\sigma)$, $c_{\text{in}}(\sigma)$, $c_{\text{noise}}(\sigma)$, and parameterization of the weight function $\lambda(\sigma)$ may refer to EDM (Karras et al., 2022).

For the latent diffusion model, we set $l = 12$ for ShapeNet and $l = 24$ for Objaverse. The final latents are obtained via 40 denoising steps. For text-conditioned 3D generation, we use CLIP to encode the input text prompts. In addition, we adopt classifier-free guidance. We randomly drop the conditioning signal with a probability of $10\%$ and set the guidance scale to 3.5 during sampling. Note that more advanced architectures (Peebles & Xie, 2022) could also be employed for the denoising network.

## B ADDITIONAL RESULTS

In Figure 8, we present additional results of text-conditioned 3D generation using our method.

Figure 9 shows the results of real-image-conditioned 3D generation. In this experiment, we simply replace the CLIP text encoder with the CLIP image encoder and train the latent diffusion model using the same dataset. Given a real image, we first apply an off-the-shelf tool to remove the background and then use the processed image as the conditional input for 3D generation. The results demonstrate that our method can still produce reasonable outcomes.

## C DISCUSSION ON CONCURRENT WORK

We discuss several concurrent works that utilize the VAE + LDM paradigm for feed-forward 3D generation. GaussianAnything (Lan et al., 2025) was concurrently developed and introduces an explicit 3D latent space to enhance interactivity. L3DG (Roessle et al., 2024) and DiffGS (Zhou et al., 2024) both design a VAE that takes 3D Gaussians as input and outputs 3D Gaussians. However, both require per-shape optimization before training the VAE, which limits scalability.

## D LIMITATIONS AND FUTURE WORK

Our method uses the vanilla 3D Gaussian representation, which is known to be challenging for extracting highly accurate geometry. Incorporating recent advancements (Huang et al., 2024; Yu et al., 2024; Guédon & Lepetit, 2023) in 3D Gaussian techniques can benefit both geometry and appearance.

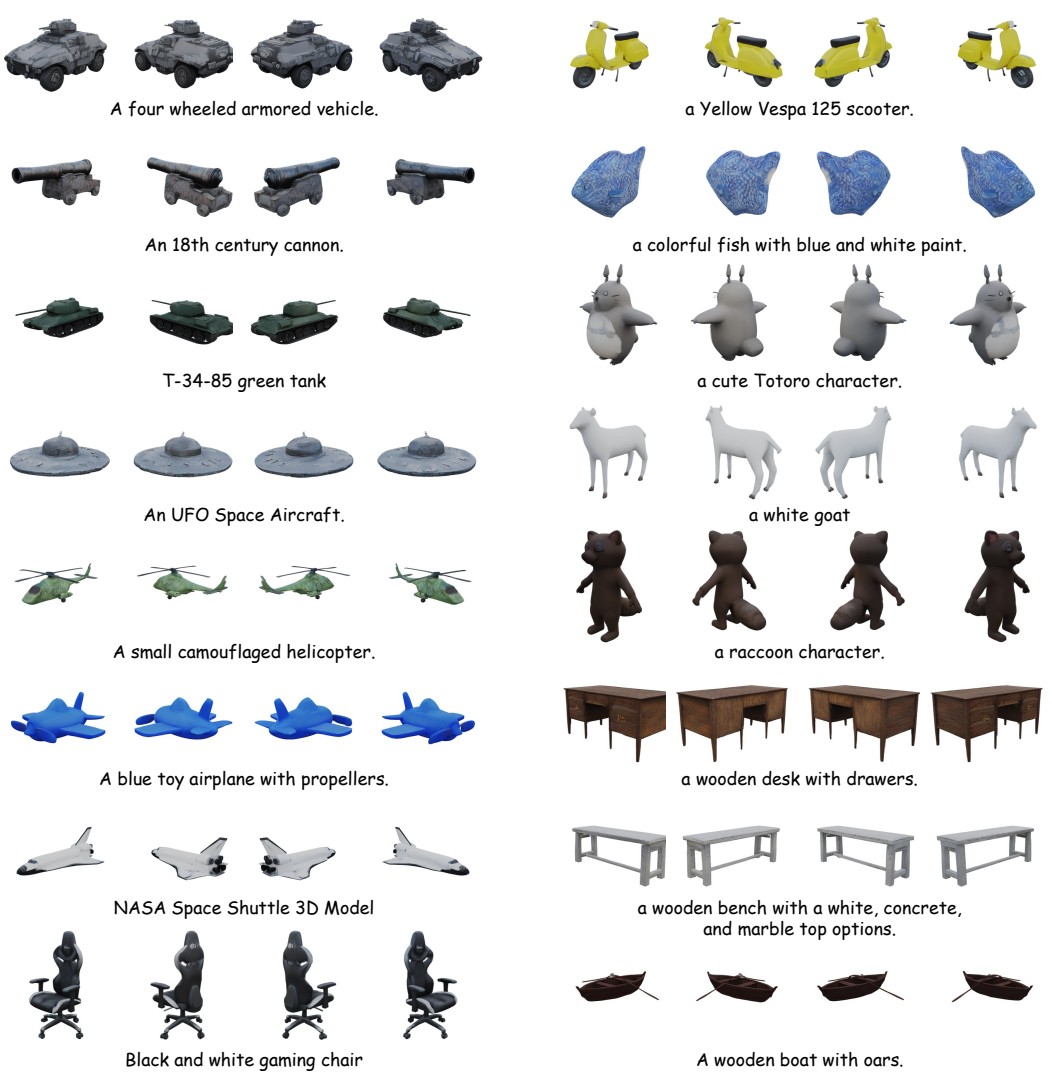

Figure 8: Additional results of text-conditioned 3D generation using our method.

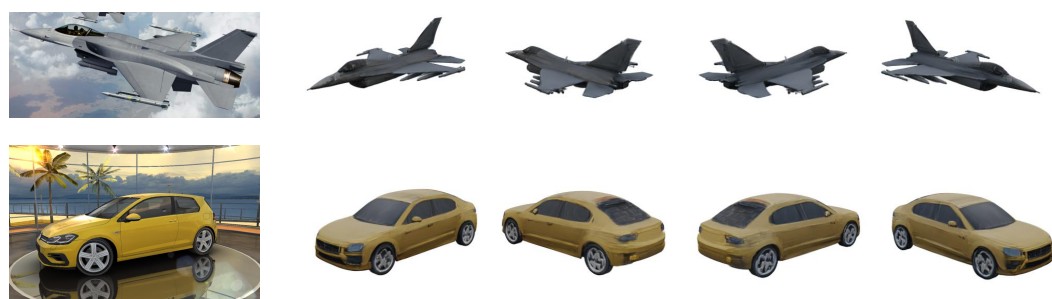

Figure 9: Image-conditioned 3D generation using our method.

