# OpenReview forum: "Atlas Gaussians Diffusion for 3D Generation"
_ICLR.cc/2025/Conference — ICLR 2025 Spotlight_

### Official Review · Reviewer_BYST · 2024-10-27

**Soundness:** 3
**Presentation:** 3
**Contribution:** 2
**Rating:** 6
**Confidence:** 3

**Summary:**

The authors proposed a representation model for generating 3D Gaussians from local patches, named Atlas Gaussians. They parameterized each patch as a sequence of feature vectors and designed a learnable function to decode 3D Gaussians, followed by the integration of UV-based sampling to generate large and theoretically infinite Gaussian points. To learn the Atlas Gaussians, the authors employed a latent diffusion model in the latent space of a VAE.

**Strengths:**

**S1.** Methodological development is clear and well presented.

**S2.** The paper is very well written and has strong motivation.

**S3.** Consideration of generation 3D Gaussians in latent diffusion model is unique and well-motivated.

**Weaknesses:**

While this paper explores a unique perspective on 3D generation, it primarily lacks proper experimental support and validation. It is difficult to discern the specific novelties and contributions (considering the three contributions outlined in L102-L107) that this paper offers compared to recent SOTAs, e.g., LN3Diff and others. Although the current methodological development is generally clear, the contributions are still not well-supported by the qualitative findings and significantly improved quantitative results reported in the paper.


**W1. Latent Diffusion Novelty:** I believe the variational diffusion model is not a novelty in terms of technical development. Besides, the authors directly utilized the EDM algorithm as their latent diffusion. What makes the integration of the latent diffusion model a unique/novel contribution other than working with a different data type such as 3D Gaussians? Please highlight specific contributions with details in the revised version.


**W2. Experimental Validation:**
*(i)* Going through the generated samples reported in the LN3Diff and this paper, it is difficult to understand the effectiveness of the proposed model. The qualitative comparison with the LN3Diff model in Fig. 5 raises concerns regarding whether the baselines have been trained to their optimal or not. Besides, for most of the pictorial depictions, I can guess that either the sampling or training has not been done properly for all the SOTA, specifically in Fig. 6. In the SOTA baselines papers, the structure of the synthesized images seems to be well preserved. Then why the authors could not achieve that? *(ii)* Did the authors run all the baselines, and try to reproduce their results? Or they just copied the FID/KID numbers from the papers? *(iii)* I would suggest reporting the mean along with std to see the robustness of the scores. *(iv)* Inference times in Tab. 2 are not comparable with GVGEN and LN3Diff as they carried out their experiments on different GPU setups. *(v)* As one of the contributions of this paper is using a transformer-based decoder, I would suggest the authors show a very brief ablation carrying out the experiments on a vanilla decoder. Also, compared to the LN3Diff which used a transformer-base decoder, what is the uniqueness of their proposed architecture?


**W3. Loss function and optimization:** According to the derived loss function, the optimization process seems to be quite complex to converge to the optimal stage as one needs control around 10 distinct losses, including the KL divergence term. I suggest the authors to shed some light onto the complete optimization process other than simply denote large number of losses with weighting parameters.


**Minor.** Please cite necessary papers that are relevant to the proposed methodology, other than just scrapping eight pages of references which includes papers that do not fall into the problem of this paper.


In its current format, I believe the paper falls below the threshold for acceptance. The score reflects an evaluation of the problem definition, motivation, and presentation. However, I am open to raising my score if the above concerns are properly addressed. Besides, I request the authors to specifically highlight the methodological contributions and novelties (compared to the current SOTA), beyond the focus on generating 3D Gaussians.

**Questions:**

Please refer to the Weaknesses section, where the primary questions have been outlined. Additional points are provided below -

1. How are the local patches parameterized as a sequence of feature vectors, and how is the learnable function designed to decode 3D Gaussians from these feature vectors?

2. Why is UV-based sampling used instead of directly generating an infinite number of 3D Gaussian points? Why is it necessary, and does it help in preserving the topology of the synthesized 3D objects?

3. How does a large number of 3D Gaussians contribute to achieving high-quality details?

4. How transformer-based decoder help to remove all the existing limitations posed by a vanilla decoder (other than the transformer decoder in LN3Diff)? Are there any strong quantitative and qualitative ablations?

5. What is the corner-encoding scheme?

---

> ### Author Response · Authors · 2024-11-21
> **Response (Part 1)**
>
> Thank you for your insightful and valuable feedback. Below are our clarifications for your concerns. We value your input and are keen to discuss any further questions or comments you may have following our response.
>
> > **W1. Latent Diffusion Novelty: I believe the variational diffusion model is not a novelty in terms of technical development. Besides, the authors directly utilized the EDM algorithm as their latent diffusion.**
>
> We do not claim that using EDM or designing a new latent diffusion model is our contribution. This is why Section 3.3 is intentionally brief, comprising only about 10 lines. As stated in L345: "Since the LDM network design is not our main contribution."
>
> Our primary contribution lies on the VAE side rather than the LDM. To apply latent diffusion to 3DGS, the critical prerequisite is designing a VAE that can efficiently map low-dimensional latents to 3DGS. This requirement is independent of the diffusion framework used (e.g., DDPM, EDM, or flow matching) or the LDM network architecture employed (e.g., those inspired by 3DShape2VecSet or DiT). To address this prerequisite, we propose:
>
> - Atlas Gaussians representation (Contribution 1, L102–103).
> - A decoder to generate Atlas Gaussians (Contribution 2, L104–105).
>
> These contributions directly address the critical prerequisite by providing a VAE capable of efficiently mapping low-dimensional latents to 3DGS, which is the core focus of our work.
>
> To our best knowledge, at the time of submission, no prior work had applied the VAE + LDM paradigm to generate 3DGS. Therefore, we claim integrating 3DGS into the VAE + LDM paradigm as our third contribution (L106–107), while explicitly not claiming LDM itself as our novelty.
>
> Our contributions have also been acknowledged by other reviewers. For example:
>
> - Reviewer 3ymM mentions: "Addressing the current challenges of VAEs in extracting efficient latent representations for generative diffusion models is important."
> - Reviewer uWac mentions: "The paper nicely integrates 3DGS with diffusion generation scheme".
> - Reviewer pZYd mentions: "Sufficient experimental results support the contributions claimed."
>
> > **What makes the integration of the latent diffusion model a unique/novel contribution other than working with a different data type such as 3D Gaussians?**
>
> The novelty lies in addressing the challenges posed by the high dimensionality of 3DGS, especially when a very large number of Gaussian points is required (e.g., 100K). Directly applying a diffusion model to such a large number of Gaussian points, instead of a compressed latent space, is highly challenging due to optimization difficulties and slower sampling.
>
> To tackle these challenges, our approach follows a common practice in generating high-dimensional data: compressing the data into a low-dimensional latent space before applying latent diffusion. This integration is crucial for enabling the efficient and scalable generation of 3DGS.

---

> > ### Author Response · Authors · 2024-11-21
> > **Response (Part 2)**
> >
> > > **W2. Experimental Validation: (i) qualitative comparison with baselines.**
> >
> > For the baseline methods in Figures 5 and 6, we directly ran the code with pretrained models provided in their official repositories, using their default settings without re-training. Importantly, as noted in L407, all text prompts used in Figure 6 are sourced from the original baseline papers. The reasons why some baselines may not perform well on certain prompts are as follows:
> >
> > - Generalization to New Text Prompts: Some methods perform well on the specific text prompts used in their own papers but fail to generalize to prompts from other baseline papers. For example, GVGEN generates reasonable results for the prompt "Pixelated model of a wooden ship with sails." (used in its original paper) but struggles with prompts like "A wooden ship with mast.", which were taken from other baselines. By using text prompts directly sourced from the original baseline papers, we ensure a fair and meaningful evaluation while highlighting these generalization issues.
> >
> > - Baseline Performance Limitations: For text prompts where the baselines succeed, the generated results are consistent with the quality reported in their respective papers. For instance, the outputs of GVGEN and Shap-E exhibit inherent blurriness, which matches the results shown in their original publications.
> >
> > - Multi-View Inconsistency: LGM, in particular, struggles with multi-view consistency when tested with more complex prompts from other baselines. In their original paper, LGM primarily showcases results for very simple text prompts, typically consisting of one or two words. When applied to more complex prompts, the generated outputs often suffer from multi-view inconsistencies, leading to noticeable artifacts in the final reconstructions.
> >
> > To validate our experiments, we first tested each baseline method with the prompts provided in their respective papers and confirmed that we could reproduce the results reported by the authors. This verification ensures that the baselines were implemented correctly before testing them with prompts from other baselines.
> >
> > > **W2. Experimental Validation: (ii) Did the authors run all the baselines, and try to reproduce their results? Or they just copied the FID/KID numbers from the papers?**
> >
> > For the ShapeNet experiment (Table 1), we only ran the baseline LN3Diff. The numbers for all other baselines were sourced from the LN3Diff paper, where the authors unified the evaluation settings across different baselines. It is important to note that variations in FID/KID implementations can significantly affect the reported numbers. To ensure a rigorous and fair comparison, we first ran the evaluation code provided by the LN3Diff authors using their generated outputs and verified that we could reproduce the reported results. We then applied the exact same evaluation code to our generated outputs to compute FID and KID, ensuring a consistent and fair comparison of all methods in Table 1.
> >
> > For the Objaverse experiment (Table 2), we carefully addressed evaluation inconsistencies. Different baseline papers used different evaluation metrics, and even when the same metric was used (e.g., CLIP Score), variations in text prompts or ground-truth datasets led to inconsistencies. To address this, we ran all the baselines ourselves and standardized the evaluation process by using the same testing prompts and evaluation code to compute the metrics. This unified approach ensures a fair comparison in Table 2.
> >
> >
> > > **W2. Experimental Validation: (iii) I would suggest reporting the mean along with std to see the robustness of the scores.**
> >
> > For the Objaverse experiment, we ran our method three times and computed the mean and standard deviation (std) of the results, as shown in the table below:
> >
> > |           | GVGEN           | LN3Diff         | LGM             | Shap-E          | Ours            |
> > | --------- | --------------- | --------------- | --------------- | --------------- | --------------- |
> > | CLIPScore | $27.41\pm 0.17$ | $27.23\pm 0.38$ | $29.60\pm 0.19$ | $29.96\pm 0.28$ | $30.53\pm 0.21$ |
> > | FID       | $132.3\pm 0.1$  | $123.9\pm 0.3$  | $116.8\pm 1.2$  | $115.1\pm 0.9$  | $110.3\pm 1.1$  |
> > | KID($\\%$) | $5.99 \pm 0.07$ | $4.46 \pm 0.07$ | $4.73 \pm 0.06$ | $4.37\pm 0.02$  | $4.03\pm 0.02$  |
> >
> > > **W2. Experimental Validation: (iv) Inference times in Tab. 2 are not comparable with GVGEN and LN3Diff as they carried out their experiments on different GPU setups.**
> >
> > The inference time of LGM, Shap-E and Ours on V100 GPU (32 GB) is 4.4s, 23.7s, and 3.6s, respectively. In Table 2, we reported results on a TITAN V GPU (12 GB) as these methods can run on it with lower memory requirements. In contrast, GVGEN and LN3Diff require more GPU memory and cannot be evaluated on TITAN V.

---

> > > ### Author Response · Authors · 2024-11-21
> > > **Response (Part 3)**
> > >
> > > > **W2. Experimental Validation: (v) I would suggest the authors show a very brief ablation carrying out the experiments on a vanilla decoder.**
> > >
> > > The decoder should take the low-dimentional latent $z_0 \in \mathbb{R}^{n\times d_0}$ as input and outputs the Atlas Gaussians representation, including geometry features $F=\\{f_i\\}\_{i=1}^M \in \mathbb{R}^{M \times \beta \times d}$, appearance features $H=\\{h_i\\}\_{i=1}^M \in \mathbb{R}^{M \times \beta \times d}$, and patch centers $X=\\{x_i\\}\_{i=1}^M \in \mathbb{R}^{M \times 3}$. Given these input and output requirements, there are many possible design choices, and we believe it is not straightforward to define what a "vanilla decoder" should be. To address your concerns, we designed a "vanilla decoder" that is as simple as possible. It is defined as follows:
> > >
> > > - Given the input latent $z_0$, we first apply a few cross-attention layers to generate shared features:  $E=\text{CrossAttn}(y, z_0)$, where $y\in\mathbb{R}^{M\times d}$ is  a learnable query initialized with a Gaussian distribution, and the output shared features are $E\in\mathbb{R}^{M\times d}$.
> > >
> > > - For the patch centers, we apply a few self-attention layers to $E$, followed by an MLP that projects the feature dimention from $d$ to $3$ to generate $X$.
> > >
> > > - Similarly, for geometry and appearance features, we use another branch consisting of a few self-attention layers applied to $E$, followed by an MLP that projects the feature dimenion from $d$ to $2\times \beta \times d$. This output is then split into two tensors $F$and $H$, each of shape $M\times \beta \times d$.
> > >
> > > - Once the Atlas Gaussians representation is obtained, we use the same UV-based sampling to generate 3DGS.
> > >
> > > We conducted this ablation study on ShapeNet Chair and evaluated the performance using PSNR. The performance of the vanilla decoder is 24.92, while our proposed method achieves 26.56. This performance gap demonstrates that our proposed techniques offer valuable improvements.
> > >
> > > > **W2. Experimental Validation: (v) compared to the LN3Diff which used a transformer-base decoder, what is the uniqueness of their proposed architecture?**
> > >
> > > The uniqueness of our architecture and LN3Diff’s lies in their tailoring to their respective inputs and outputs. Our method uses a latent set representation (following 3DShape2VecSet) as input and generates the Atlas Gaussians representation as output, while LN3Diff uses triplane-based inputs and outputs triplane features. These fundamental differences naturally lead to distinct decoder designs.
> > >
> > > It is also worth noting that LN3Diff does not employ a "vanilla decoder" but incorporates specialized design choices tailored to its triplane-based pipeline. Similarly, our decoder is specifically designed to align with the Atlas Gaussians framework and handle set-based representations effectively.
> > >
> > > > **W3. Loss function and optimization**
> > >
> > > Our network training utilizes three types of loss to train the VAE:
> > >
> > > - KL loss ($\mathcal{L}_\text{KL}$): A standard loss in VAE.
> > >
> > > - Rendering-related loss ($\mathcal{L}\_\text{render}$, $\mathcal{L}\_\text{LPIPS}$): These losses supervise the rendered RGB, alpha, and depth outputs. Additionally, we employ the LPIPS loss to enhance visual fidelity (L330). Such losses are standard in learning 3DGS and have been widely used in works like Lan et al. (2024), Xu et al. (2024a), and Zou et al. (2023).
> > >
> > > - Geometry-related loss ($\mathcal{L}\_\text{center}$, $\mathcal{L}\_\mu$): We encourage patches to distribute uniformly across the surface (L320) and align Gaussian locations with the ground-truth surface. This strategy provides better initialization and makes optimization easier, which follows existing methods like TriplaneGaussian (Zou et al., 2023).
> > >
> > > > **Minor. Please cite necessary papers that are relevant to the proposed methodology, other than just scrapping eight pages of references which includes papers that do not fall into the problem of this paper.**
> > >
> > > While we have carefully curated references to ensure that "each section of Section 2: Related Work clearly connects to the proposed method" (as noted by Reviewer pZYd), we sincerely welcome your feedback. If there are specific references you believe are less relevant, we would be happy to revisit and refine them to better align with the focus of our paper.
> > >
> > > > **How are the local patches parameterized as a sequence of feature vectors, and how is the learnable function designed to decode 3D Gaussians from these feature vectors?**
> > >
> > > Each local patch is parameterized with a patch center, geometry features, and appearance features. For example, the geometry features are represented as $\mathbf{f}\_{i}=(\mathbf{f}\_{i1}, \mathbf{f}\_{i2}, \mathbf{f}\_{i3}, \mathbf{f}\_{i4})\in \mathbb{R}^{4\times d}$ (L190) ,where $\mathbf{f}_{i}$ forms a sequence of length four. This parameterization also applies to the appearance features.
> > >
> > > The design of the learnable function is explicitly provided in Equations (2) and (5) of the main paper.

---

> > > > ### Author Response · Authors · 2024-11-21
> > > > **Response (Part 4)**
> > > >
> > > > > **Why is UV-based sampling used instead of directly generating an infinite number of 3D Gaussian points? Why is it necessary, and does it help in preserving the topology of the synthesized 3D objects?**
> > > >
> > > > Directly representing or generating an infinite number of 3DGS is highly impractical. Even in our method, we only claim "theoretically infinite" to emphasize the flexibility of the representation.
> > > >
> > > > In practice, it is unnecessary to generate an infinite number of 3DGS. A sufficiently large number of 3DGS is typically adequate for achieving high-quality rendering (L72). UV-based sampling is used to efficiently generate this large but finite set of 3DGS. However, this approach does not directly help in preserving the topology of the synthesized 3D objects.
> > > >
> > > > > **How does a large number of 3D Gaussians contribute to achieving high-quality details?**
> > > >
> > > > A single 3D Gaussian has limited representational capacity and can be viewed as a localized sampling of the underlying signal. By increasing the number of Gaussians, we achieve a denser sampling of the shape, enabling our model to capture fine-grained geometric details and high-frequency features more effectively.
> > > >
> > > > > **How transformer-based decoder help to remove all the existing limitations posed by a vanilla decoder (other than the transformer decoder in LN3Diff)? Are there any strong quantitative and qualitative ablations?**
> > > >
> > > > We respectfully clarify that our paper does not claim that the transformer-based decoder removes all existing limitations posed by a vanilla decoder. As mentioned in previous responses, defining what constitutes a "vanilla decoder" is subjective given our specific input and output requirements. To address this, we designed a simpler network as a "vanilla decoder" and provided an ablation study to evaluate its performance, as outlined in our earlier comments.
> > > >
> > > > > **What is the corner-encoding scheme?**
> > > >
> > > > As mentioned in L189–190, "we parameterize the geometry and appearance features as the features at the four corners of the local patch in the UV space." The corner-encoding scheme refers to encoding features that represent the information specific to each corner of the local patch. We will clarify this further in the final version.

---

> > > > > ### Comment · Reviewer_BYST · 2024-11-25
> > > > > **Official Comment by Reviewer BYST**
> > > > >
> > > > > I read the rebuttal carefully and thank the authors. I greatly appreciate the authors for clarifying some of my concerns regarding technical novelty and experimentation.
> > > > >
> > > > > While I still believe that the methodological development reflects the latent diffusion model, EDM, 3dshape2vecset as one of the highlights, I feel that, as a reader, this detracts from the key part of the technical contribution, which is the Atlas Gaussian representation.
> > > > >
> > > > > Second, I am still unclear about the complete optimization process of the total loss functions, which includes around ten distinct loss functions with different weighting parameters. Did the authors optimize the losses alternately or altogether? A brief explanation of the optimization details would greatly help readers reproduce and understand the intuition behind optimizing all these functions together.
> > > > >
> > > > > Third and most importantly, I am still unsure about the LN3Diff visualization results reported in Fig. 5. Compared to the LN3Diff paper's generated visualizations, the reported images seem distorted. What are the reasons for achieving such images running on the same datasets?
> > > > >
> > > > >
> > > > > I would request the authors to briefly justify the LN3Diff results and the optimization process, if possible. With that being said, I believe the paper establishes its foundation clearly enough to be above the threshold. Hence, I am raising the initial rating by one point.

---

> > > > > > ### Author Response · Authors · 2024-11-25
> > > > > > **Response to Reviewer BYST**
> > > > > >
> > > > > > Thank you again for your valuable feedback.
> > > > > >
> > > > > > > **While I still believe that the methodological development reflects the latent diffusion model, EDM, 3dshape2vecset as one of the highlights, I feel that, as a reader, this detracts from the key part of the technical contribution, which is the Atlas Gaussian representation.**
> > > > > >
> > > > > > In the methodological development, we include EDM and the latent diffusion model from 3DShape2VecSet in Section 3.3 (approximately 10 lines) primarily for completeness, as the paper focuses on 3D generation. We explicitly state that LDM is not our main contribution and keep this section concise.
> > > > > >
> > > > > > One potential alternative could be removing Section 3.3 (approximately 10 lines)  entirely and omitting any mention of LDM. However, we believe it is important to retain this brief section for self-containment and completeness, especially considering that LDM is indispensable in the VAE + LDM paradigm.
> > > > > >
> > > > > > We sincerely welcome any feedback and are open to revisiting and refining this section based on specific suggestions to improve the balance and emphasis in the paper.
> > > > > >
> > > > > > > **Did the authors optimize the losses alternately or altogether? A brief explanation of the optimization details would greatly help readers reproduce and understand the intuition behind optimizing all these functions together.**
> > > > > >
> > > > > > As mentioned in the rebuttal, all the losses used in our optimization are quite standard and have been widely employed in existing works, particularly in 3DGS learning. Regarding your concern about "different weighting parameters", as noted in L970, we set $\lambda_r=1$, meaning all loss weights are 1 except for $\lambda_{KL}=1e^{-4}$, which simplifies the weighting scheme considerably.
> > > > > >
> > > > > > We have already provided the optimization details in L969–L973. As described, "The VAE training consists of two stages":
> > > > > >
> > > > > > - "In the first stage, $\lambda_r$ in Equation (11) is set to 0" (L969). This means only the KL loss and the geometry-related loss are active. In this stage, the predicted Gaussian locations are aligned with the ground-truth surface, providing better initialization for learning 3DGS.
> > > > > >
> > > > > > - As mentioned, "Rendering occurs only in the second stage with $\lambda_r$ set to 1" (L970). In the second stage, all losses are active and contribute to the optimization.
> > > > > >
> > > > > > This two-stage training strategy also follows existing methods like TriplaneGaussian (Zou et al., 2023).
> > > > > >
> > > > > > > **What are the reasons for achieving such images running on the same datasets?**
> > > > > >
> > > > > > As mentioned in the rebuttal, for quantitative evaluation, we ensured a rigorous and fair comparison by running the evaluation code provided by the LN3Diff authors on both their generated results and our results. The results in Table 1 demonstrate that our method outperforms LN3Diff in these quantitative metrics.
> > > > > >
> > > > > > We believe the most important metrics are quantitative metrics, under which our method outperforms LN3Diff by notable margins. Regarding qualitative results, we observed that the examples shown in prior work were often selected to highlight the advantages of each method. We realized this issue when running the codes of baseline methods (we will release our code, which is in the supplementary material).
> > > > > >
> > > > > > For the visual comparison between our approach and LN3Diff, we found that a significant portion of the results are comparable. In many cases, the results of LN3Diff are as good as those shown in their original paper. However, the instances where our approach performs better than LN3Diff are more than the ones where LN3Diff performs better than ours. This aligns with the quantitative improvements reported in Table 1. Naturally, we showed the ones where ours are better. Again, we emphasize that the fairest comparisons are based on the quantitative metrics.

---

### Official Review · Reviewer_uWac · 2024-11-04

**Soundness:** 3
**Presentation:** 4
**Contribution:** 3
**Rating:** 8
**Confidence:** 3

**Summary:**

The paper introduces a new framework called Atlas Gaussians Diffusion for 3D shape generation. It models the shape as a union of local patches as 3D Gaussians defined in UV space. This allows the generation process to generate theoretically unlimited number of points while capturing more underlying details of the 3D shape. Extensive quantitative and qualitative experiments demonstrate their results are better than the current SOTA.

**Strengths:**

The paper nicely integrates 3DGS with diffusion generation scheme and achieves state of the art results in 3D shape generation.

The use of Atlas Gaussians to represent 3D shapes as union of local patch makes it capable of capturing better details and higher scalability.

Decomposing the 3D shape generation into local Gaussian patches and using UV-based sampling to handle shapes of varying complexity.

The evaluation is thorough with reasonable qualitative and quantitative results and achieves SOTA results in 3D shape generation.

**Weaknesses:**

While in 4.4 discuss about memory consumption and # of patches, the paper is lacking thorough comparison on inference times (esp. with the increase of # of 3DGS) to further justify the value of the framework.

How could the current approach address gap for even finer details of the shapes as Gaussian representation struggles in fine details and often results in blurry effect. Is it only a matter of the # of Gaussians or patches?

The author mentioned "We take inspiration from the literature that 3D-based neural networks learn better 3D features for recognition and outperform multi-view based networks that leverage massive 2D labeled data." without any citation. There are arguments on both sides about overall performance with both approaches. This sentence seems to be subjective.

**Questions:**

Do the authors plan to open source the model?

Can UV-based sampling potentially generate artifacts? (e.g. inconsistency or distortions)

How well does the proposed framework handle noisy or sparse data?

---

> ### Author Response · Authors · 2024-11-21
> **Response (Part 1)**
>
> Thank you for your insightful and valuable feedback. Below are our clarifications for your concerns.
>
> > **Comparison on inference times with the increase of # of 3DGS.**
>
> We provide the computational times (measured on a V100 GPU) with an increasing number of 3DGS (#3DGS) in the table below. As #3DGS increases, the computational overhead primarily comes from decoding more 3DGS in the decoder and rendering them. During inference, the denoising time in latent diffusion remains constant, while the decoder and rendering time increases as #3DGS increases. However, compared to the denoising time, the computational overhead from using more 3DGS is negligible.
>
> | #3DGS $\approx$                              | 8K      | 32K     | 100K    |
> | -------------------------------------------- | -------:| -------:| -------:|
> | Denoising in Inference (Per Shape)           | 3.4 s   | 3.4 s   | 3.4 s   |
> | Decoder + Rendering in Inference (Per Shape) | 0.050 s | 0.062 s | 0.087 s |
>
> > **How could the current approach address gap for even finer details of the shapes as Gaussian representation struggles in fine details and often results in blurry effect. Is it only a matter of the # of Gaussians or patches?**
>
> Increasing the number of 3DGS could contribute to capturing finer details. However, it is not solely a matter of the number of 3DGS. We find that optimization, specifically how the network is trained, also plays a crucial role. For example, aligning the locations of 3DGS to the surface (as described in L320) provides a better initialization and facilitates the learning of finer details. This is why we adopt a two-stage training strategy (L969), following (Xu et al., 2024a) and (Zou et al., 2023).
>
> That said, efficiently aligning a large number of 3DGS (e.g., 100k) to the surface remains a challenging problem. As a workaround, we currently sample a subset of points and compute CD/EMD on this subset ( $\mathcal{L}_\mu(S)$ ). With increased computational resources and more efficient loss functions for better initializing the 3DGS locations, achieving finer details could become more feasible.
>
> > **The author mentioned "We take inspiration from the literature that 3D-based neural networks learn better 3D features for recognition and outperform multi-view based networks that leverage massive 2D labeled data." without any citation. There are arguments on both sides about overall performance with both approaches. This sentence seems to be subjective.**
>
> Thank you for pointing this out. Our statement is inspired by the development of 3D recognition tasks in the deep learning era, and we acknowledge that there are ongoing debates regarding the performance of different approaches.
> Taking 3D shape classification as an example, early methods used multi-view representations as input for 3D shape classification tasks [1]. Subsequently, 3D-based neural networks were introduced and demonstrated better performance, as seen in methods like PointNet [2] for point clouds. A chronological overview and comparison of multi-view-based networks versus 3D-based networks can be found in Figure 2 of the survey [3]. For other 3D recognition tasks, such as 3D object detection and 3D semantic segmentation, detailed comparisons are provided in Figures 7 and 10 of the same survey [3].
> We will include these references in the final version to make the statement more balanced and objective.
>
> [1] Multi-view Convolutional Neural Networks for 3D Shape Recognition
>
> [2] PointNet: Deep Learning on Point Sets for 3D Classification and Segmentation
>
> [3] Deep Learning for 3D Point Clouds: A Survey

---

> ### Author Response · Authors · 2024-11-21
> **Response (Part 2)**
>
> > **Do the authors plan to open source the model?**
>
> Yes, we have included the code in the supplementary material and commit to open-sourcing the model.
>
> > **Can UV-based sampling potentially generate artifacts? (e.g. inconsistency or distortions)**
>
> Artifacts could potentially arise due to inconsistencies across patches, as mentioned in L319: "Note that patches may overlap, similar to AtlasNet." However, during rendering, we only consider the union of all 3D Gaussian points decoded from the patches. As long as this union globally matches the ground truth images, local inconsistencies across patches do not affect the overall representation. Empirically, we observe that our native 3D representation generalizes quite well.
>
> > **How well does the proposed framework handle noisy or sparse data?**
>
> For multi-view inputs, our method can handle noisy camera poses because the poses of the input images are not fixed. We train our VAE directly on renderings from the G-buffer Objaverse (L361), where the elevation of the input image views ranges from 5° to 30° (excluding bottom views). This variability demonstrates the robustness of our approach to pose noise.
>
> For point cloud inputs, we conducted an ablation study using the ShapeNet Chair dataset and evaluated the PSNR of the VAE. As shown in the table below, we added Gaussian random noise to the input point cloud coordinates with a distribution of $\mathcal{N}(0, \sigma^2)$. When the noise scale is small ($\sigma=0.001$), the performance remains almost unchanged. However, as the noise scale increases ($\sigma = 0.01$), the performance decreases more significantly. Additionally, we performed experiments using sparse input point clouds with only 1024 points. The results demonstrate that our framework is capable of handling sparser data.
>
> |      | Ours  | $\sigma=0.001$ | $\sigma=0.01$ | $\vert \mathcal{P} \vert = 1024$ |
> | ---- | :-----: | :--------------: | :-------------: | :--------------------------------: |
> | PSNR | 26.56 | 26.52          | 25.71         | 26.50                            |

---

### Official Review · Reviewer_pZYd · 2024-11-04

**Soundness:** 3
**Presentation:** 2
**Contribution:** 3
**Rating:** 8
**Confidence:** 4

**Summary:**

The authors introduce Atlas Gaussian, a 3D generation approach grounded in diffusion models.

First, they train a VAE to map 3D shapes into a compact latent space, which is then decoded through a two-branch structure into patch centers and features. These patch centers and features can subsequently be decoded into 3D Gaussian distributions.
Next, they train a Latent Diffusion Model (LDM) within the learned latent space to serve as the generation model.

Atlas Gaussian offers three main advantages:
1. Using UV-based sampling within a unit square, it can generate a large quantity of 3D Gaussians efficiently.
2. It incorporates a nonlinear, learnable weight function through an MLP-projected positional encoding, enhancing representational power compared to traditional linear interpolation weights.
3. It is computationally efficient with minimal memory usage.

These benefits are substantiated by comprehensive experiments and ablation studies.

**Strengths:**

1. I appreciate how the introduction section clearly explains and flows through the limitations of recent works, outlining how the authors address these challenges with their proposed method.

2. I appreciate how each section of Section 2: Related Work clearly connects to the proposed method.

3. Sufficient experimental results support the contributions claimed.

4. I value the release of code which enhances transparency and reproducibility.

**Weaknesses:**

1. I suspect the term “atlas” is intended in the same sense as in AtlasNet (L125~128). If so, adding an explanation would be helpful, as it is part of the method’s name but is not clarified in the text.

2. The method section is presented in a list-like format, making it difficult to read smoothly.

3. The clarity of the writing is lacking. Although the text contains many details, the explanations are somewhat vague, requiring readers to infer and interpret a great deal.

4. Some notations lack sufficient explanation. Although readers might deduce their meanings through context, clearer descriptions would be beneficial. (Also see questions below.)

5. Figure 3 lacks detail and does not clearly relate to the main text, particularly Section 3.2, Stage 1: VAE. For instance,

(1) The steps from $\bar{z} \rightarrow z_0$ are ambiguous. Is the process

$z' = \mathnormal{CrossAttn}(\bar{z}, \mathcal{P})$

$z'' = \mathnormal{CrossAttn}(z', \mathcal{I})$

$z_0 = \mathnormal{MLP}(z'')$

($z'$ and $z''$ notations arbitrarily assigned as they are not explicitly defined)

correct?

(2) My understanding is that $z_0$ of Figure 3 represents the end of the VAE encoder, while subsequent steps are part of the decoder. It would help to indicate these details explicitly.

6. The conditioning ($\mathcal{C}$ of Section 3.3) mechanism lacks detail; although some information is provided in the supplementary material, it remains insufficient.

7. An ablation study should be conducted to analyze the impact of the loss hyperparameters ($\lambda_r$ and $\lambda_{KL}$).

**Questions:**

1. L060 "However, these approaches often focus solely on modeling geometry without considering the appearance attributes."
Can you give a more detailed explanation?

2. What does $d$ represent in L189 and L135 (Equation 4)?

3. What does $n$ represent in L256 and L268?

4. What is the exact relationship between the shape information $\mathcal{S}$ (L263) and the point cloud $\mathcal{P}$ as well as the sparse-view RGB images $\mathcal{I}$?
From my understanding, the feature dimension of $\mathcal{P}$ is $\mathbb{R}^{|\mathcal{P}| \times d}$, and the feature dimension of $\mathcal{I}$ is $\mathbb{R}^{(|\mathcal{I}| \times h \times w) \times d}$.
How, precisely, are these features of $\mathcal{P}$ and $\mathcal{I}$, and the feature dimensions related to $\mathcal{S}$, particularly in mathematical terms?

5. How is the learnable query $y$ of the VAE decoder (L277, L284) initialized?

---

> ### Author Response · Authors · 2024-11-21
> **Response (Part 1)**
>
> Thank you for your insightful and valuable feedback. While three other reviewers have rated our presentation as either "excellent" or "good,", we acknowledge that there is always room for improvement. We are grateful for your constructive suggestions and are committed to addressing your concerns to enhance the clarity and presentation of our paper. Below, we provide clarifications and responses to your comments.
>
> > **I suspect the term “atlas” is intended in the same sense as in AtlasNet (L125~128).**
>
> Yes. The term "atlas" has the same meaning as in AtlasNet. As a concept in mathematics, an atlas is a collection of charts that collectively cover a manifold. We will add the explanation in the final version.
>
> > **Figure 3 lacks detail and does not clearly relate to the main text, particularly Section 3.2, Stage 1: VAE. For instance, (1) ... (2) ...**
>
> **(1) Is the process $z' = \text{CrossAttn}(\bar{z}, \mathcal{P}),
> z'' = \text{CrossAttn}(z', \mathcal{I}),
> z_0 = \text{MLP}(z'')$ correct?**
>
> Yes, that is correct. We will clarify this point and add the explanation in the final version.
>
> **(2) My understanding is that $z_0$  of Figure 3 represents the end of the VAE encoder, while subsequent steps are part of the decoder.**
>
> Yes, that is correct. As mentioned in the paper:
>
> - L256: "Encoder. The encoder takes shape information as input and outputs a latent set $z_0$"
>
> - L272: "Decoder. The decoder recovers patch features ... from the latent code $z_0$"
>
> - L227: " The latent $z_0$ is used for latent diffusion."
>
>   We will clarify this further in the final version of the paper.
>
> > **The conditioning ( $\mathcal{C}$ of Section 3.3) mechanism lacks detail**
>
> The conditioning $\mathcal{C}$ is represented as a vector:
>
> - For unconditional generation, $\mathcal{C}$ is set as a learnable parameter, implemented as `torch.nn.Embedding(1, d)`.
> - For text-conditioned generation (conditional generation), $\mathcal{C}$ is set as the CLIP embedding of the input text prompts, which is also a vector.
>
> In Equation (12) of the main paper, $\mathcal{C}$ serves as the key/value in the cross-attention mechanism. We will provide additional details in the final version to make this clearer.
>
> > **An ablation study should be conducted to analyze the impact of the loss hyperparameters ($\lambda_{r}$ and $\lambda_{KL}$).**
>
> Regarding $\lambda_r$, as mentioned in the supplementary material (L972), doubling or halving $\lambda_r$ results in almost identical loss curves. For simplicity, we set $\lambda_r=1$.
>
> To analyze the effect of $\lambda_{KL}$, we conducted an ablation study using the ShapeNet Plane dataset and evaluated the PSNR of the VAE. As shown in the table, decreasing $\lambda_{KL}$ leads to improved performance. However, in the VAE + LDM paradigm, the KL loss plays an important role in preventing arbitrarily high-variance latent spaces. Balancing these factors, we set $\lambda_{KL}=1e^{-4}$ for simplicity.
>
> | $\lambda_{KL}$ | $1e^{-3}$ | $1e^{-4}$ | $2e^{-5}$ |
> | -------------- | --------- | --------- | --------- |
> | PSNR           | 28.76     | 29.59     | 29.85     |
>
> > **L060 "However, these approaches often focus solely on modeling geometry without considering the appearance attributes." Can you give a more detailed explanation?**
>
> In many previous works, the focus has been on generating representations of the geometric surface, such as 3D point clouds or iso-surfaces. These methods do not generate the colors or textures of the 3D objects, meaning the appearance attributes are not considered. For example, LION generates only 3D points, while 3DShape2VecSet generates occupancy fields. These representations inherently lack color information.

---

> ### Author Response · Authors · 2024-11-21
> **Response (Part 2)**
>
> > **What does $d$ represent in L189 and L135 (Equation 4)?**
>
> $d$ is the feature dimension of $\mathbf{f}\_{ij}$ and  $\mathbf{h}\_{ij}$, where $j=\{1, 2, 3, 4\}$. For example, $\mathbf{f}\_{i1}\in \mathbb{R}^d$, $\mathbf{f}\_{i2}\in \mathbb{R}^d$, $\mathbf{f}\_{i3}\in \mathbb{R}^d$,  $\mathbf{f}\_{i4}\in \mathbb{R}^d$. Thus, $\mathbf{f}\_{i}=(\mathbf{f}\_{i1}, \mathbf{f}\_{i2}, \mathbf{f}\_{i3}, \mathbf{f}\_{i4})\in \mathbb{R}^{4\times d}$ . This definition is also consistent with Equation (4).
>
>
> > **What does $n$ represent in L256 and L268?**
>
> $n$ represents the size of the set in the latent set representation. As mentioned in L257, the encoder ouputs  $\mathbf{z}_0$, which is a latent set representation first proposed by 3DShape2VecSet. Intuitively, a vanilla VAE uses a single latent vector of dimension $d_0$ to define the latent space. In contrast, the latent set representation defines the latent space using a set of latent vectors, where each vector is of dimension $d_0$, and the size of the set is $n$. Therefore, the latent set $\mathbf{z}_0\in\mathbb{R}^{n\times d_0}$ . We will clarify this further in the final version.
>
> > **What is the exact relationship between the shape information $\mathcal{S}$ (L263) and the point cloud $\mathcal{P}$ as well as the sparse-view RGB images $\mathcal{I}$?**
>
> The shape information $\mathcal{S}$ includes both the point cloud $\mathcal{P}$ and the sparse-view RGB images $\mathcal{I}$. In terms of the mathematical formula, you are correct in the previous comments: $z' = \text{CrossAttn}(\bar{z}, \mathcal{P}),
> z'' = \text{CrossAttn}(z', \mathcal{I}),
> z_0 = \text{MLP}(z'')$. We will clarify this further in the final version.
>
> > **How is the learnable query $\mathbf{y}$ of the VAE decoder (L277, L284) initialized?**
>
> The learnable query $\mathbf{y}$ is initialized using a Gaussian distribution, implemented as: `torch.nn.Parameter(torch.randn(n, d) * 0.02)`.

---

> > ### Comment · Reviewer_pZYd · 2024-12-02
> >
> > I have carefully reviewed all the comments from other reviewers and the authors' rebuttals. I would like to thank the authors for addressing all of my concerns and questions. However, I believe it is still not sufficient to raise the score to an 8, so I will maintain my current score. I am looking forward to seeing the final paper incorporate all the reviewers' concerns and improve the overall flow of the writing.

---

### Official Review · Reviewer_3ymM · 2024-11-07

**Soundness:** 3
**Presentation:** 4
**Contribution:** 3
**Rating:** 8
**Confidence:** 3

**Summary:**

This paper introduces Atlas Gaussians, a new latent representation learning framework designed to improve the efficiency of 3D object generation by addressing the current challenges faced by VAEs. By modeling 3D shapes as a union of local patches, the authors develop a transformer-based decoder to map the low-dimensional latent representations of each patch via UV-based sampling and promote time efficiency when generating large samples of 3D Gaussians. Additionally, the introduced Atlas Gaussian representation considers a disentangled learning of geometric and appearance features, resulting in faster convergence.

**Strengths:**

- The paper is well-written and organized; hence it is easy for readers to follow.

- Addressing the current challenges of VAEs in extracting efficient latent representations for generative diffusion models is important.

- The introduction of the Atlas Gaussians representation with disentangled learning of geometry and appearance features of 3D shapes is novel.

**Weaknesses:**

- The proposed approach includes both the Atlas Gaussian representation with disentangled geometry and appearance learning mechanism, as well as a transformer-based encoder. It is unclear which component contributes more significantly to the overall performance.

**Questions:**

- It is unclear how geometry and appearance features are disentangled in the latent space. While the loss function is designed to learn each patch’s center and appearance features through supervision, the process for learning geometric features remains vague.

---

> ### Author Response · Authors · 2024-11-21
>
> Thank you for your insightful and valuable feedback. Below are our clarifications for your concerns.
>
> > **The proposed approach includes both the Atlas Gaussian representation with disentangled geometry and appearance learning mechanism, as well as a transformer-based encoder. It is unclear which component contributes more significantly to the overall performance.**
>
> To ensure I understand your comment correctly, let me rephrase it: suppose we have a hypothetical baseline VAE capable of generating 3DGS. Adding our **transformer-based encoder** to this baseline would result in a performance gain of $\Delta_1$​, while adding our **Atlas Gaussian representation** would result in a performance gain of $\Delta_2$. Are you asking which of these gains is larger, i.e., which component contributes more significantly to the overall performance?"
>
> Unfortunately, there is no existing baseline VAE using 3DGS at the time of our submission. Without such a baseline as a reference, it is challenging to quantitatively compare the individual contributions of these components in isolation.
>
> Based on our best interpretation of your comment, this question may reflect a concern about whether all the new components in our VAE are necessary for achieving the reported performance. We believe both components are indeed crucial, though we currently lack direct quantitative measures for $\Delta_1$​ and $\Delta_2$​. However, we have evidence supporting the necessity and effectiveness of each component individually, i.e. both $\Delta_1 > 0$ and $\Delta_2 > 0$:
>
> - Transformer-Based Encoder:
>   Our design follows the approach in 3DShape2VecSet, leveraging a latent set representation. The usefulness of this design has been demonstrated through the ablation studies in the original 3DShape2VecSet paper. We build on their proven techniques to enhance our model.
>
> - Atlas Gaussian Representation:
>   We provide an ablation study in Table 4 in the main paper to validate its effectiveness. The study compares the full Atlas Gaussian design with simpler alternatives (e.g., linear weights, no disentanglement, no global feature, etc.), showing that the full design outperforms these naive baselines. This supports our claim that the representation is a necessary and effective improvement.
>
> > **It is unclear how geometry and appearance features are disentangled in the latent space.**
>
> The geometry and appearance features refer to $\mathbf{f}_i$ and $\mathbf{h}_i$, respectively, as defined in Lines 188–189 of the paper. These features are not defined in the latent space. The disentanglement of geometry and texture features is achieved through the use of separate branches to learn them (L105, L516).
>
> As mentioned in L520, both geometry and appearance features are generated from the shared latent $\mathbf{z}_0$. This means the latent space is not explicitly divided into two subspaces, where one corresponds to geometry and the other to appearance. Instead, the disentanglement is performed during feature generation. We will clarify this point further in the final version.
>
> > **The process for learning geometric features remains vague.**
>
> As shown in Equation (2) of the main paper, the geometry features $\mathbf{f}_i$ are used to predict the Gaussian location $\mu$. These Gaussian location $\mu$ are supervised through Equation (9), which regularizes $\mu$ to align with the GT surface. During end-to-end training, the geometry features $\mathbf{f}_i$ will also receive gradients, enabling their learning.

---

### Comment · Area_Chair_muBV · 2024-11-24
**Discussion with the authors will conclude soon!!**

Dear Reviewers,

The discussion with the authors will conclude soon. The authors have provided detailed rebuttals. If there are any points that you feel have not been adequately clarified or if there are misunderstandings in their responses, please take this opportunity to raise them now. Thank you for your contributions to this review process.

---

### Meta-Review · Area_Chair_muBV · 2024-12-21

**Metareview:**

This paper proposes a novel framework, Atlas Gaussians Diffusion, for 3D shape generation. The method represents 3D shapes as a union of local patches defined as 3D Gaussians in UV space, enabling the generation of theoretically unlimited points while preserving finer details. The integration of the 3D Gaussian Splatting (3DGS) method with a diffusion generation scheme results in a scalable and detailed shape generation process. Extensive quantitative and qualitative experiments validate the framework's superiority, achieving state-of-the-art (SOTA) performance in 3D shape generation.

Reviewers unanimously praised the paper, highlighting the innovative integration of 3D Gaussian Splatting with diffusion-based generation.
The use of Atlas Gaussians to represent shapes can also effectively capturefine details and ensuring scalability.
Comprehensive evaluations with strong quantitative and qualitative results that establish the framework's SOTA performance.

The paper introduces a well-validated, innovative approach to 3D shape generation, combining methodological novelty with strong experimental results. The unanimous positive feedback supports its acceptance as a significant contribution to the field.

**Additional Comments On Reviewer Discussion:**

While reviewers initially raised concerns, these were effectively addressed during the rebuttal, resulting in increased ratings and unanimous support.

---

### Decision · Program_Chairs · 2025-01-22

Accept (Spotlight)